# EFO$_k$-CQA: Towards Knowledge Graph Complex Query Answering beyond Set Operation

## Abstract

To answer complex queries on knowledge graphs, logical reasoning over incomplete knowledge is required due to the open-world assumption. Learning-based methods are essential because they are capable of generalizing over unobserved knowledge. Therefore, an appropriate dataset is fundamental to both obtaining and evaluating such methods under this paradigm. In this paper, we propose a comprehensive framework for data generation, model training, and method evaluation that covers the combinatorial space of Existential First-order Queries with multiple variables (EFO$_k$). The combinatorial query space in our framework significantly extends those defined by set operations in the existing literature. Additionally, we construct a dataset, EFO$_k$-CQA, with 741 query types for empirical evaluation, and our benchmark results provide new insights into how query hardness affects the results. Furthermore, we demonstrate that the existing dataset construction process is systematically biased and hinders the appropriate development of query-answering methods, highlighting the importance of our work. Our code and data are provided in `https://anonymous.4open.science/r/EFOK-CQA/README.md`.

## 1 Introduction

The Knowledge Graph (KG) is a powerful database that encodes relational knowledge into a graph representation (Vrandečić & Krötzsch, 2014; Suchanek et al., 2007), supporting downstream tasks (Zhou et al., 2007; Ehrlinger & Wöß, 2016) with essential factual knowledge. However, KGs suffer from incompleteness during its construction (Vrandečić & Krötzsch, 2014; Carlson et al., 2010), which is formally acknowledged as Open World Assumption (OWA) (Libkin & Sirangelo, 2009). The task of Complex Query Answering (CQA) proposed recently has attracted much research interest (Hamilton et al., 2018; Ren & Leskovec, 2020). This task ambitiously aims to answer database-level complex queries described by logical complex connectives (conjunction $\land$, disjunction $\lor$, and negation $\neg$) and quantifiers[1] (existential $\exists$) (Wang et al., 2022; Ren et al., 2023; Leskovec, 2023). Currently, learning-based methods dominate the CQA tasks because they can empirically generalize to unseen knowledge as well as prevent the resource-demanding symbolic search.

The thriving of learning-based methods also puts an urgent request on high-quality benchmarking methods, including datasets with comprehensive coverage of queries and sound answers, and fair evaluation protocol for learning-based approaches. In the previous study, datasets are developed by progressively expanding the **syntactical expressiveness**, where conjunction (Hamilton et al., 2018), union (Ren et al., 2020), negation (Ren & Leskovec, 2020), and other operators (Liu et al., 2021) are taken into account sequentially. In particular, the dataset proposed in Ren & Leskovec (2020) contains all logical connectives and becomes the standard training set for model development. Wang et al. (2021) proposed a large evaluation benchmark EFO-1-QA that systematically evaluates the combinatorial generalizability of CQA models on such queries. More related works are included in Appendix A.

However, the queries in aforementioned datasets (Ren & Leskovec, 2020; Wang et al., 2021) are recently justified as "Tree-Form" queries (Yin et al., 2023) as they rely on the tree combinations

---

[1]The universal quantifier is usually not considered in query answering tasks, as a common practice from both CQA on KG (Wang et al., 2022; Ren et al., 2023) and database query answering (Poess & Floyd, 2000).

of set operations. Compared to the well-established TPC-H decision support benchmark (Poess & Floyd, 2000) for database query processing, queries in existing CQA benchmarks (Ren & Leskovec, 2020; Wang et al., 2021) have two common shortcomings: (1) lack of **combinatorial answers**: only one variable is queried, and (2) lack of **structural hardness**: all existing queries subject to the structure-based tractability (Rossi et al., 2006; Yin et al., 2023). It is rather questionable whether existing CQA data under such limited scope can support the future development of methodologies for general decision support with open-world knowledge.

The goal of this paper is to establish a new framework that addresses the aforementioned shortcomings to support further research in complex query answering on knowledge graphs. Our framework is formally motivated by the well-established investigation of constraint satisfaction problems, in which all queries can be formulated. In general, the contribution of our work is four folds.

**Complete coverage** We capture the complete Existential First Order (EFO) queries from their rigorous definitions, underscoring both **combinatorial hardness** and **structural hardness** and extending the existing coverage (Wang et al., 2021) which covers only a subset of $EFO_1$ query. The captured query family is denoted as $EFO_k$ where $k$ stands for multiple variables.

**Curated datasets** We derive $EFO_k$-CQA dataset, a enormous extension of the previous EFO-1-QA benchmark (Wang et al., 2021) and contains 741 types of query. We design several rules to guarantee that our dataset includes high-quality nontrivial queries, particularly those that contain multiple query variables and are not structure-based tractable.

**Convenient implementation** We implement the entire pipeline for query generation, answer sampling, model training and inference, and evaluation for the undiscussed scenarios of **combinatorial answers**. Our pipeline is backward compatible, which supports both set operation-based methods and more recent ones.

**Results and findings** We evaluate six representative CQA methods on our benchmark. Our results refresh the previous empirical findings and further reveal the structural bias of previous data.

## 2 PROBLEM DEFINITION

### 2.1 EXISTENTIAL FIRST ORDER (EFO) QUERIES ON KNOWLEDGE GRAPHS

Given a set $\mathcal{E}$ of entities and a set $\mathcal{R}$ of relations, a knowledge graph $\mathcal{KG}$ encodes knowledge as a set of factual triple $\mathcal{KG} = \{(h, r, t)\} \subset \mathcal{E} \times \mathcal{R} \times \mathcal{E}$. According to the OWA, the knowledge graph that we have observed $\mathcal{KG}_o$ is only part of the real knowledge graph, meaning that $\mathcal{KG}_o \subset \mathcal{KG}$.

The existing research only focuses on the logical formulas without universal quantifiers (Ren et al., 2023; Wang et al., 2023). We then offer the definition of it based on strict first order logic.

**Definition 1** (Term). *A term is either a variable $x$ or an entity $a \in \mathcal{E}$.*

**Definition 2** (Atomic formula). *$\phi$ is an atomic formula if $\phi = r(h, t)$, where $r \in \mathcal{R}$ is a relation, $h$ and $t$ are two terms.*

**Definition 3** (Existential first order formula). *The set of the existential formulas is the smallest set $\Phi$ that satisfies the following[2]:*

*(i) For atomic formula $r(h, t)$, itself and its negation $r(h, t), \neg r(h, t) \in \Phi$*
*(ii) If $\phi, \psi \in \Phi$, then $(\phi \wedge \psi), (\phi \vee \psi) \in \Phi$*
*(iii) If $\phi \in \Phi$ and $x_i$ is any variable, then $\exists x_i \phi \in \Phi$.*

**Definition 4** (Free variable). *If a variable $y$ is not associated with a quantifier, it is called a free variable, otherwise, it is called a bounded variable. We write $\phi(y_1, \cdots, y_k)$ to indicate $y_1, \cdots, y_k$ are the free variables of $\phi$.*

**Definition 5** (Sentence and query). *A formula $\phi$ is a sentence if it contains no free variable, otherwise, it is called a query. In this paper, we always consider formula with free variable, thus, we use formula and query interchangeably.*

**Definition 6** (Substitution). *For $a_1, \cdots, a_k$, where $a_i \in \mathcal{E}$, we write $\phi(a_1/y_1, \cdots, a_k/y_k)$ or simply $\phi(a_1, \cdots, a_k)$ for the result of simultaneously replacing all free occurrence of $y_i$ in $\phi$ by $a_i$, $i = 1, \cdots, k$.*

---

[2]We always assume all variables are named differently as common practice in logic.

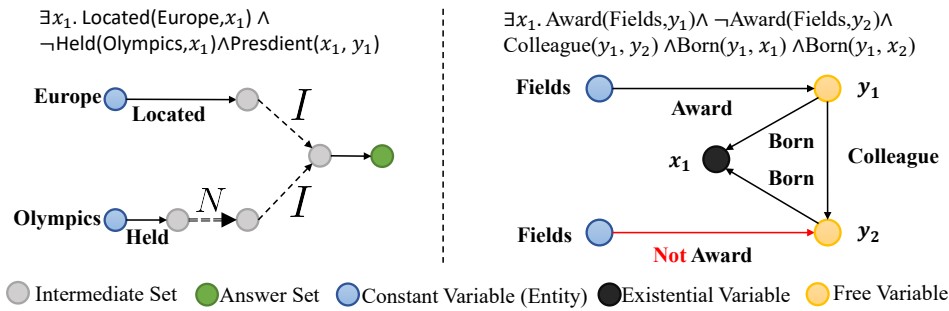

Figure 1: Operator Tree versus Query Graph. **Left**: An operator tree representing a given query "List the presidents of European countries that have never held the Olympics" (Ren & Leskovec, 2020); **Right**: A query graph representing a given query "Find a pair of persons who are both colleagues and co-authors and were born in the same country, with one having awarded the fields medal while the another not", which is both a multigraph and a cyclic graph, containing two free variables.

**Definition 7** (Answer of an EFO query). *For a given existential query $\phi(y_1, \cdots, y_k)$ and a knowledge graph $\mathcal{KG}$, its answer is a set that defined by*

$$\mathcal{A}[\phi(y_1, \cdots, y_k)] = \{(a_1, \cdots, a_k))|a_i \in \mathcal{E}, i = 1, \cdots, k, \phi(a_1, \cdots, a_k) \text{ is True in } \mathcal{KG}\}.$$

**Definition 8** (Disjunctive Normal Form (DNF)). *For any existential formula $\phi(y_1, \cdots, y_k)$, it can be converted to the Disjunctive normal form as shown below:*

$$\phi(y_1, \cdots, y_k) = \gamma_1(y_1, \cdots, y_k) \vee \cdots \vee \gamma_m(y_1, \cdots, y_k), \tag{1}$$
$$\gamma_i(y_1, \cdots, y_k) = \exists x_1, \cdots, x_n.\rho_{i1} \wedge \cdots \wedge \rho_{it}, \tag{2}$$

where $\rho_{ij}$ is either an atomic formula or the negation of it, $x_i$ is called an existential variable.

DNF form has a strong property that $\mathcal{A}[\phi(y_1, \cdots, y_k)] = \cup_{i=1}^m \mathcal{A}[\gamma_i(y_1, \cdots, y_k)]$, which allows us to only consider conjunctive formulas $\gamma_i$ and then aggregate those answers to retrieve the final answers. This practical technique has been used in many previous research (Long et al., 2022; Ren et al., 2023). Therefore, we only discuss conjunctive formulas in the rest of this paper.

## 2.2 Constraint satisfaction problem for EFO queries

Formally, a constraint satisfaction problem (CSP) $\mathcal{P}$ can be represented by a triple $\mathcal{P} = (X, D, C)$ where $X = (x_1, \cdots, x_n)$ is an $n$-tuple of variables, $D = (D_1, \cdots, D_n)$ is the corresponding $n$-tuple of domains, $C = (C_1, \cdots, C_t)$ is $t$-tuple constraint, each constraint $C_i$ is a pair of $(S_i, R_{S_i})$ where $S_i$ is a set of variables $S_i = \{x_{i_j}\}$ and $R_{S_i}$ is the constraint over those variables (Rossi et al., 2006).

Historically, there are strong parallels between CSP and conjunctive queries in knowledge bases (Gottlob et al., 1999; Kolaitis & Vardi, 1998). The terms correspond to the variable set $X$. The domain $D_i$ of a constant entity contains only itself, while it is the whole entity set $\mathcal{E}$ for other variables. Each constraint $C_i$ is binary that is induced by an atomic formula or its negation, for example, for an atomic formula $r(h, t)$, we have $S_i = \{h, t\}$, $R_{S_i} = \{(h, t)|h, t \in \mathcal{E}, (h, r, t) \in \mathcal{KG}\}$. Finally, by the definition of existential quantifier, we only consider the answer of free variable, rather than tracking all terms within the existential formulas.

**Definition 9** (CSP answer of conjunctive formula). *For a conjunctive formula $\gamma$ in Equation 2 with $k$ free variables and $n$ existential variables, the answer set of it formulated as CSP instance is:*

$$\overline{\mathcal{A}}[\gamma(y_1, \cdots, y_k)] = \mathcal{A}[\gamma^\star(y_1, \cdots, y_{n+k})], \text{ where } \gamma^\star = \rho_{i1} \wedge \cdots \wedge \rho_{it}.$$

This shows that the inference of existential formulas is easier than solving CSP instances since the existential variables do not need to be kept track of.

## 2.3 The representation of query

To give an explicit representation of existential formula, Hamilton et al. (2018) firstly proposes to represent a formula by operator tree, where each node represents the answer set for a sub-query, and the logic operators in it naturally represent set operations. This method allows for the recursive computation from constant entity to the final answer set in a bottom-up manner (Ren & Leskovec,

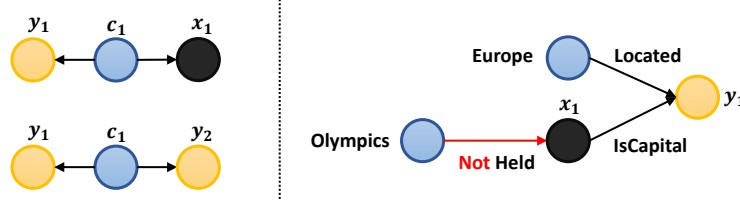

Figure 2: Left: Example of trivial abstract query graph, in the upper left graph, the $x_1$ is redundant violating Assumption 13, in the bottom left graph, answers for the whole query can be decomposed to answer two free variables $y_1$ and $y_2$ alone, violating Assumption 14. Right: Example of new query graph that is not included in previous benchmark (Wang et al., 2021) even though it can be represented by operator-tree. The representation of query graph follows Figure 1.

2020). We also provide full details of the operator tree and tree-form query in Appendix C. However, this representation method is inherently directed, acyclic, and simple, therefore more recent research breaks these constraints by being bidirectional (Liu et al., 2022; Wang et al., 2022) or being cyclic or multi (Yin et al., 2023). To meet these new requirements, they propose to represent the formula by the query graph (Yin et al., 2023), which inherits the convention of constraint network in representing CSP instance. We utilize this design and further extend it to represent $\text{EFO}_k$ formula that contains multiple free variables. We provide the illustration and comparison of the operator tree and the query graph in Figure 1, where we show the strong expressiveness of the query graph. We also provide the formal definition of query graph as follows:

**Definition 10** (Query graph). *Let $\gamma$ be a conjunctive formula in equation 2, its query graph is defined by $G(\gamma) = \{(h, r, t, \{T/F\})\}$, where an atomic formula $\rho = r(h, t)$ in $\gamma$ corresponds to $(h, r, t, T)$ and $\rho = \neg r(h, t)$ corresponds to $(h, r, t, F)$.*

Therefore, any conjunctive formulas can be represented by a query graph, in the rest of the paper, we use query graphs and conjunctive formulas interchangeably.

## 3   THE COMBINATORIAL SPACE OF $\text{EFO}_k$ QUERIES

Although previous research has given a systematic investigation in the combinatorial space of operator trees (Wang et al., 2021), the combinatorial space of the query graph is much more challenging due to the extremely large search space and the lack of explicit recursive formulation. To tackle this issue on a strong theoretical background, we put forward additional assumptions to exclude trivial query graphs. Such assumptions or restrictions also exist in the previous dataset and benchmark (Ren & Leskovec, 2020; Wang et al., 2021). Specifically, we propose to split the task of generating data into two levels, the abstract level, and the grounded level. At the abstract level, we create *abstract query graph*, at the grounded level, we provide the abstract query graph with the relation and constant and instantiate it as a query graph. In this section, we elaborate on how we investigate the scope of the nontrivial $\text{EFO}_k$ query of interest step by step.

### 3.1   NONTRIVIAL ABSTRACT QUERY GRAPH OF $\text{EFO}_k$

The abstract query graph is the ungrounded query graph without information of certain knowledge graphs, and we give an example in Figure 3.

**Definition 11** (Abstract query graph). *The abstract query graph $\mathcal{G} = (V, E, f, g)$ is a directed graph with three node types, {**Constant Entity, Existential Variable, Free variable**}, and two edge types, {**positive, negative**}. The $V$ is the set of nodes, $E$ is the set of directed edges, $f$ is the function maps node to node type, $g$ is the function maps edge to edge type.*

**Definition 12** (Grounding). *For an abstract query graph $\mathcal{G}$, a grounding is a function $I$ that maps it into a query graph $I(\mathcal{G})$.*

We propose two assumptions of the abstract query graph as follows:

**Assumption 13** (No redundancy). *For a abstract query graph $\mathcal{G}$, there is not a subgraph $\mathcal{G}_s \subsetneq \mathcal{G}$ such that for every grounding $I$, $\mathcal{A}[I(\mathcal{G})] = \mathcal{A}[I(\mathcal{G}_s)]$.*

**Assumption 14** (No decomposition). *For an abstract query graph $\mathcal{G}$, there are no such two subgraphs $\mathcal{G}_1$, $\mathcal{G}_2$, satisfying that $\mathcal{G}_1, \mathcal{G}_2 \subsetneq \mathcal{G}$, such that for every instantiation $I$, $\mathcal{A}[I(\mathcal{G})] = \mathcal{A}[I(\mathcal{G}_1)] \times \mathcal{A}[I(\mathcal{G}_2)]$, where the $\times$ represents the Cartesian product.*

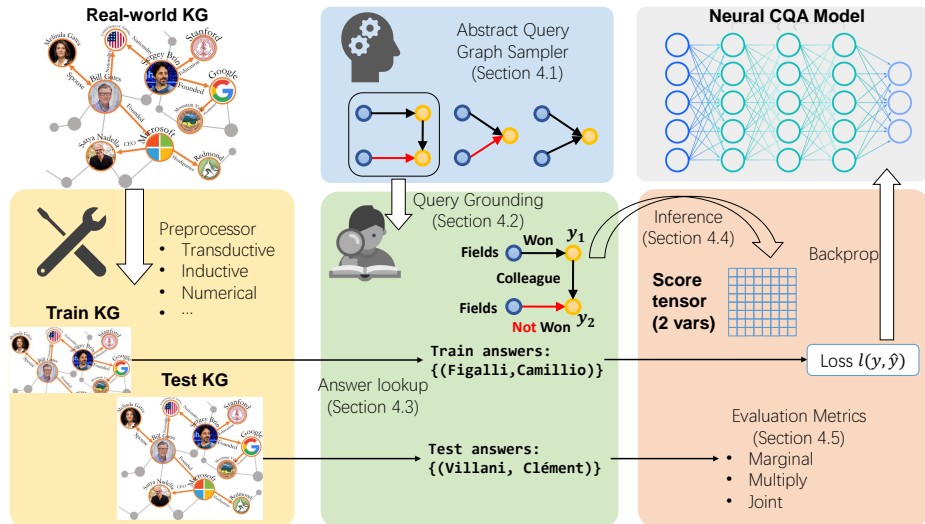

Figure 3: Illustration of the all functionalities of our framework. Real-world KG is preprocessed and fed into our pipeline, which contains the whole process of data generation and supports end-to-end machine learning as well as evaluation. Additionally, the figure of the real-world KG is taken from `https://medium.com/@fakrami/re-evaluation-of-knowledge-graph-completion-methods-7dfe2e981a77`.

The assumption 14 inherits the idea of the **structural** decomposition technique in CSP (Gottlob et al., 2000), which allows for solving a CSP instance by solving several sub-problems and combining the answer together based on topology property. Additionally, meeting these two assumptions in the grounded query graph is extremely computationally costly thus we avoid it in practice.

We provide some easy examples to be excluded for violating the assumptions above in Figure 2.

## 3.2 Nontrivial query graph of $\text{EFO}_k$

Similarly, we propose two assumptions on the query graph.

**Assumption 15** (Meaningful negation). *For any negative edge $e$ in query graph $G$, we require removing it results in different CSP answers: $\overline{\mathcal{A}}[G - e] \neq \overline{\mathcal{A}}[G]$.*[3]

Assumption 15 treats negation separately because of the fact that for any $\mathcal{KG}$, any relation $r \in \mathcal{R}$, there is $|\{(h,t)|h, t \in \mathcal{E}, (h, r, t) \in \mathcal{KG}\}| \ll |\mathcal{E}|^2$, which means that the constraint induced by the negation of an atomic formula is much less "strict" than the one induced by a positive atomic formula.

**Assumption 16** (Appropriate answer size). *There is a constant $M \ll |\mathcal{E}|$ to bound the candidate set for each free variable $f_i$ in $G$, such that for any $i$, $|\{a_i \in \mathcal{E}|(a_1, \cdots, a_i, \cdots, a_k) \in \mathcal{A}[G]\}| \leqslant M$.*

We note the Assumption 16 **extends** the "bounded negation" assumption in the previous dataset (Ren & Leskovec, 2020; Wang et al., 2021). We give an example "Find a city that is located in Europe and is the capital of a country that has not held the Olympics" in Figure 2, where the candidate set of $x_1$ is in fact bounded by its relation with the $y_1$ variable but not from the bottom "Olympics" constant, hence, this query is excluded in their dataset due to the directionality of operator tree.

Overall, the scope of the formula investigated in this paper surpasses the previous EFO-1-QA benchmark because of: (1). We include the $\text{EFO}_k$ formula with multiple free variables for the first time; (2). We include the whole family of $\text{EFO}_1$ query, many of them can not be represented by operator tree; (3) Our assumption is more systematic than previous ones as shown by the example in Figure 2. More details are offered in Appendix D.3.

## 4 Framework

We develop a versatile framework that supports five key functionalities fundamental to the whole CQA task: (1) Enumeration of nontrivial abstract query graphs as discussed in Section 3; (2) Sample

---

[3]Ideally, we should expect them to have different answers as the existential formulas, however, this is computation costly and difficult to sample in practice, which is further discussed in Appendix D.

grounding for the abstract query graph; (3) Compute answer for any query graph efficiently; (4) Support implementation of existing CQA models; (5) Conduct evaluation including newly introduced $\text{EFO}_k$ queries with multiple free variables. We explain each functionality in the following. An illustration of the first three functionalities is given in Figure 3, where we show how each functionality cooperates to help CQA tasks. We note that preprocessing allows us to extend our framework to more avant-garde settings, like inductive settings or graphs with numerics, more discussions in Appendix G.

### 4.1 ENUMERATE ABSTRACT QUERY GRAPH

As discussed in Section 3, we are able to abide by those assumptions as well as **enumerate** all possible query graphs within a given search space where certain parameters, including the number of constants, free variables, existential variables, and the number of edges are all given, shown in Figure 3. Additionally, we apply the graph isomorphism algorithm to avoid duplicated query graphs being generated. More details for our generation method are provided in Appendix D.1.

### 4.2 GROUND ABSTRACT QUERY GRAPH

To ground an abstract query graph $\mathcal{G}$ and comply with the assumption 15, we split the abstract query graph into two parts, the positive part and the negative part, $\mathcal{G} = \mathcal{G}_p \cup \mathcal{G}_n$. Then the grounding process is also split into two steps: 1. Sample grounding for the positive subgraph $\mathcal{G}_p$ and compute its answer, 2. Ground the $\mathcal{G}_n$ to decrease the answer got in the first step. Details in Appendix D.2.

Finally, to fulfill the assumption 16, we follow the previous practice of manually filtering out queries that have more than 100 answers (Ren & Leskovec, 2020; Wang et al., 2021), as we have introduced the $\text{EFO}_k$ queries, we slightly soften this constraint to be no more than $100 \times k$ answers.

### 4.3 ANSWER FOR EXISTENTIAL FORMULA

As illustrated in Section 2.2, the answer to an existential formula can be solved by a CSP solver, however, we also show in Definition 9 that solve it as CSP leads to huge computation costs. Thus, we develop our own algorithm following the standard solving technique of CSP, which ensures consistency conditions in the first step, and do the backtracking to get the final answers in the second step. Finally, we select part of our sampled queries and double-check it with the CSP solver `https://github.com/python-constraint/python-constraint`.

### 4.4 LEARNING-BASED METHODS

As the query graph is an extension to the operator tree regarding the express ability to existential formulas, we are able to reproduce CQA models that are initially implemented by the operator tree in our new framework. Specifically, since the operator tree is directed and acyclic, we compute its topology ordering that allows for step-by-step computation in the query graph. This algorithm is illustrated in detail in the Appendix F. Therefore, our pipeline is backward compatible.

Conversely, for the newly proposed models that are based on query graphs, the original operator tree framework is not able to implement them, while our framework is powerful enough. We have therefore clearly shown that the query graph representation is more powerful than the previous operator tree and is able to support arbitrary existential formulas as explained in Section 2.3.

### 4.5 EVALUATION PROTOCOL

As we have mentioned in Section 2.1, there is an observed knowledge graph $\mathcal{KG}_o$ and a full knowledge graph $\mathcal{KG}$. Thus, there is a set of observed answers $\mathcal{A}_o$ and a set of full answers $\mathcal{A}$ correspondingly. Since the goal of CQA is to tackle the challenge of OWA, it has been a common practice to evaluate CQA models by the "hard" answers $\mathcal{A}_h = \mathcal{A} - \mathcal{A}_o$ (Ren et al., 2020; 2023). However, to the best of our knowledge, there has not been a systematic evaluation protocol for $\text{EFO}_k$ queries, thus we leverage this idea and propose three types of different metrics to fill the research gap in the area of evaluation of queries with multiple free variables, and thus have combinatorial answers.

**Marginal.** For any free variable $f_i$, its full answer is $\mathcal{A}^{f_i} = \{a_i \in \mathcal{E} | (a_1, \cdots, a_i, \cdots, a_k) \in \mathcal{A}\}$, the observed answer of it $\mathcal{A}_o^{f_i}$ is defined similarly. This is termed "solution projection" in CSP

Table 1: HIT@10 scores(%) for inferring queries with one free variable on FB15k-237. We denote $e$, $c$ as the number of existential variables, constant entities correspondingly. SDAG represents Simple Directed Acyclic Graph, Multi for multigraph, and Cyclic for cyclic graph. AVG.($c$) and AVG.($e$) is the average score of queries with the number of constant entities / existential variables fixed.

| Model | $c$ \ $e$ | 0 | 1 | | 2 | | | AVG.($c$) | AVG. |
|---|---|---|---|---|---|---|---|---|---|
| | | SDAG | SDAG | Multi | SDAG | Multi | Cyclic | | |
| BetaE | 1 | 31.4 | 33.0 | 22.3 | 21.1 | 17.7 | 30.7 | 22.1 | |
| | 2 | 57.2 | 36.2 | 35.5 | 29.3 | 29.4 | 45.3 | 32.5 | 36.4 |
| | 3 | 80.0 | 53.1 | 53.6 | 38.2 | 37.8 | 58.2 | 42.1 | |
| | AVG.($e$) | 59.3 | 43.8 | 40.6 | 33.8 | 32.7 | 49.3 | | |
| LogicE | 1 | 34.4 | 34.9 | 23.0 | 21.4 | 17.4 | 30.3 | 22.4 | |
| | 2 | 60.0 | 38.4 | 36.8 | 29.8 | 29.3 | 45.3 | 33.0 | 36.7 |
| | 3 | 83.0 | 55.5 | 55.5 | 38.5 | 37.8 | 57.8 | 42.4 | |
| | AVG.($e$) | 62.2 | 46.0 | 42.0 | 34.2 | 32.6 | 49.1 | | |
| ConE | 1 | 34.9 | 35.4 | 23.6 | 21.8 | 18.4 | 34.2 | 23.5 | |
| | 2 | 61.0 | 39.1 | 38.4 | 32.0 | 31.5 | 50.2 | 35.2 | 39.0 |
| | 3 | 84.8 | 56.7 | 57.1 | 41.1 | 40.0 | 63.4 | 44.9 | |
| | AVG.($e$) | 63.4 | 47.0 | 43.5 | 36.5 | 34.7 | 54.1 | | |
| CQD | 1 | **39.0** | 34.2 | 17.6 | 17.4 | 12.7 | 28.7 | 18.7 | |
| | 2 | 50.7 | 33.8 | 33.6 | 28.4 | 28.4 | 45.7 | 31.4 | 35.9 |
| | 3 | 58.4 | 49.6 | 52.4 | 39.3 | 39.1 | 60.4 | 42.6 | |
| | AVG.($e$) | 50.7 | 41.4 | 38.4 | 33.8 | 32.4 | 50.2 | | |
| LMPNN | 1 | 38.6 | 37.8 | 21.8 | 22.9 | 17.8 | 31.7 | 23.2 | |
| | 2 | 62.2 | 40.2 | 35.0 | 30.8 | 28.1 | 44.4 | 32.5 | 35.8 |
| | 3 | 86.6 | 56.9 | 51.9 | 38.3 | 35.3 | 55.8 | 40.8 | |
| | AVG.($e$) | 65.4 | 47.8 | 39.6 | 34.5 | 30.8 | 48.0 | | |
| FIT | 1 | 38.7 | **42.7** | **32.5** | **26.1** | **22.5** | **41.5** | **28.8** | |
| | 2 | **65.5** | **47.7** | **48.2** | **39.7** | **40.1** | **56.5** | **43.4** | **47.0** |
| | 3 | **84.2** | **63.9** | **63.5** | **50.5** | **50.4** | **63.5** | **53.6** | |
| | AVG.($e$) | **65.8** | **54.7** | **51.5** | **44.9** | **43.7** | **57.5** | | |

theory (Greco & Scarcello, 2013) to evaluate whether the locally retrieved answer can be extended to an answer for the whole problem. Then, we rank the hard answer $\mathcal{A}_h^{f_i} = \mathcal{A}^{f_i} - \mathcal{A}_o^{f_i}$[4], against those non-answers $\mathcal{E} - \mathcal{A}^{f_i} - \mathcal{A}_o^{f_i}$ and use the ranking to compute standard metrics like MRR, HIT@K for every free variable. Finally, the metric on the whole query graph is taken as the average of the metric on all free variables. We note that this metric is an extension of the previous design proposed by Liu et al. (2021). However, this metric has the inherent drawback that it fails to evaluate the combinatorial answer by the $k$-length tuple and thus fails to find the correspondence among free variables.

**Multiply.** Because of the limitation of the marginal metric discussed above, we propose to evaluate the combinatorial answer by each $k$-length tuple $(a_1, \cdots, a_k)$ in the hard answer set $\mathcal{A}_h$. Specifically, we rank each $a_i$ in the corresponding node $f_i$ the same as the marginal metric. Then, we propose the HIT@$n^k$ metric, it is 1 if all $a_i$ is ranked in the top $n$ in the corresponding node $f_i$, and 0 otherwise.

**Joint.** Finally, we note these metrics above are not the standard way of evaluation, which is based on a joint ranking for all the $\mathcal{E}^k$ combinations of the entire search space. We propose to estimate the joint ranking in a closed form given certain assumptions, see Appendix E for the proof and details.

## 5 THE EFO$_k$-CQA DATASET AND BENCHMARK RESULTS

### 5.1 THE EFO$_k$-CQA DATASET

With the help of our framework developed in Section 4, we are able to develop a new dataset called EFO$_k$-CQA, whose combinatorial space is parameterized by the number of constants, existential and free variables, and the number of edges. EFO$_k$-CQA dataset includes 741 different abstract query graphs in total.

Then, we conduct experiments on our new EFO$_k$-CQA dataset with six representative CQA models including BetaE (Ren & Leskovec, 2020), LogicE (Luus et al., 2021), and ConE (Zhang et al., 2021), which are built on the operator tree, CQD (Arakelyan et al., 2020), LMPNN (Wang et al., 2023), and

---

[4]We note $\mathcal{A}_h^{f_i}$ can be empty for some free variable or even for all free variables, making these marginal metrics not reliable, details in Appendix E.

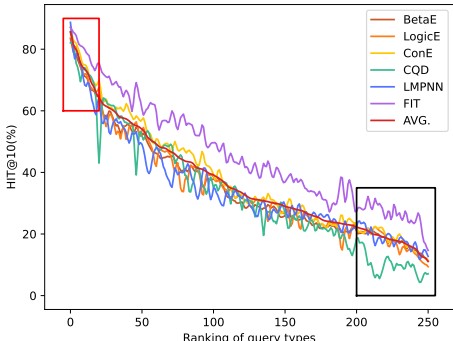

Figure 4: Relative performance of the six representative CQA models in referring queries with one free variable, where the ranking of query types is determined by the average HIT@10 score. A Gaussian filter with sigma=1 is added to smooth the curve. We also use the red box to highlight the easiest queries and the black box to highlight the most challenging ones.

FIT (Yin et al., 2023) which are built on query graph. The experiments are conducted in two parts, (1). the queries with one free variable, specifically, including those that can not be represented by operator tree; (2). the queries that contain multiple free variables.

The parameters and the generation process, as well as its statistics, are detailed in Appendix D.4, where we also provide a dataset constructed in inductive settings. However, we mainly focus on transductive settings in the main paper since there are very few inductive models to benchmark.

We have made some adaptations to the implementation of CQA models, allowing them to infer $EFO_k$ queries, full detail is offered in Appendix F. The experiment is conducted on a standard knowledge graph FB15k-237 (Toutanova & Chen, 2015) and additional experiments on other standard knowledge graphs FB15k and NELL are presented in Appendix H.

## 5.2 BENCHMARK RESULTS FOR $k = 1$

Because of the great number of abstract query graphs, we follow Wang et al. (2021) to group query graphs by three factors: (1). the number of constant entities; (2). the number of existential variables, and (3). the topology of the query graph[5]. The result is shown in Table 1 and Figure 4.

**Structure analysis.** Firstly, we find a clear monotonic trend that adding constant entities makes a query easier while adding existing variables makes a query harder, which the previous research (Wang et al., 2021) fails to uncover. Besides, we are the first to consider the topology of query graphs: when the number of constants and existential variables is fixed, we have found the originally investigated queries that correspond to Simple Directed Acyclic Graphs (SDAG) are generally easier than the multigraphs ones but harder than the cyclic graph ones. This is an intriguing result that greatly deviates from traditional CSP theory in close world which finds that the cyclic graph is NP-complete, while the acyclic graph is tractable (Carbonnel & Cooper, 2016). Our conjecture for this intriguing result in the open world is that the cyclic graph contains one more constraint than SDAG that serves as a source of information for CQA models, while the multigraph tightens an existing constraint and thus makes the query harder.

**Model analysis.** For models that are built on operator tree, including BetaE, LogicE, and ConE, their relative performance is steady among all breakdowns and is consistent with their reported score in the original dataset (Ren & Leskovec, 2020), showing similar generalizability. However, for models that are built on query graphs, including CQD, LMPNN, and FIT, we found that LMPNN performs generally better than CQD in SDAG, but falls behind CQD in multigraphs and cyclic graphs. We assume the reason is that LMPNN requires training while CQD does not, however, the original dataset are **biased** which only considers SDAG, leading to the result that LMPNN doesn't generalize well to the unseen tasks with different topology property. We expect future CQA models may use our framework to address this issue of biased data and generalize better to more complex queries.

Moreover, by the detailed observation in Figure 4, we plot two boxes, the red one and the black one. In the red box, we find that even the worst model and the best model have pretty similar performance

---

[5]To facilitate our discussion, we make a further constraint in our $EFO_k$-CQA dataset that the total edge is at most as many as the number of nodes, thus, a graph can not be both a multigraph and a cyclic graph.

Table 2: HIT@10 scores(%) of three different types for answering queries with two free variables on FB15k-237. The constant number is fixed to be two. $e$ is the number of existential variables. The SDAG, Multi, and Cyclic are the same as Table 1.

| Model | HIT@10 Type | $e = 0$ | | $e = 1$ | | | $e = 2$ | | | AVG. |
|---|---|---|---|---|---|---|---|---|---|---|
| | | SDAG | Multi | SDAG | Multi | Cyclic | SDAG | Multi | Cyclic | |
| BetaE | Marginal | 54.5 | 50.2 | 49.5 | 46.0 | 58.8 | 37.2 | 35.5 | 58.3 | 43.8 |
| | Multiply | 27.3 | 22.4 | 22.3 | 16.9 | 26.2 | 16.9 | 13.9 | 25.7 | 18.3 |
| | Joint | 6.3 | 5.4 | 5.2 | 4.2 | 10.8 | 2.2 | 2.3 | 9.5 | 4.5 |
| LogicE | Marginal | 58.2 | 50.9 | 52.2 | 47.4 | 60.4 | 37.7 | 35.8 | 59.2 | 44.6 |
| | Multiply | 32.1 | 23.1 | 24.9 | 18.1 | 28.3 | 18.1 | 14.8 | 26.6 | 19.5 |
| | Joint | 6.8 | 6.0 | 6.1 | 4.5 | 12.3 | 2.5 | 2.7 | 10.3 | 5.1 |
| ConE | Marginal | 60.3 | 53.8 | 54.2 | 50.3 | **66.2** | 40.1 | 38.5 | **63.7** | 47.7 |
| | Multiply | 33.7 | 25.2 | 26.1 | 19.8 | 32.1 | 19.5 | 16.3 | 30.3 | 21.5 |
| | Joint | 6.7 | 6.4 | 6.2 | 4.8 | 12.6 | 2.6 | 2.7 | 10.9 | 5.3 |
| CQD | Marginal | 50.4 | 46.5 | 49.1 | 45.6 | 59.7 | 33.5 | 33.1 | 61.5 | 42.8 |
| | Multiply | 28.9 | 23.4 | 25.4 | 19.5 | 31.3 | 17.8 | 16.0 | 30.5 | 21.0 |
| | Joint | **8.0** | 8.0 | 7.4 | 6.0 | **13.9** | 3.6 | 3.9 | **12.0** | **6.4** |
| LMPNN | Marginal | 58.4 | 51.1 | 54.9 | 49.2 | 64.7 | 39.6 | 36.1 | 58.7 | 45.4 |
| | Multiply | 35.0 | 26.7 | 29.2 | 21.7 | **33.4** | 21.4 | 17.0 | 28.4 | 22.2 |
| | Joint | 7.6 | 7.5 | 7.1 | 5.3 | 12.9 | 2.8 | 2.9 | 9.5 | 5.2 |
| FIT | Marginal | **64.3** | **61.0** | **63.1** | **60.7** | 58.5 | **49.0** | **49.1** | 60.2 | **54.3** |
| | Multiply | **39.7** | **32.2** | **35.9** | **27.8** | 27.4 | **29.5** | **26.8** | **32.4** | **29.2** |
| | Joint | 7.4 | **9.0** | **7.8** | **6.5** | 10.1 | 3.7 | **4.6** | 10.6 | **6.4** |

in these easiest queries despite that they may differ greatly in other queries. In the black box, we note that CQD (Arakelyan et al., 2020), though designed in a rather general form, is pretty unstable when comes to empirical evaluation, as it has a clear downward curve and deviates from other model's performance enormously in most difficult query types. Therefore, though its performance is better than LMPNN and comparable to BetaE on average as reported in Table 1, its unsteady performance suggests its inherent weakness, especially when the users are risk-sensitive and desire a trustworthy machine-learning model that does not crash in extreme cases (Varshney, 2019).

We note FIT is designed to infer all $EFO_1$ queries and is indeed able to outperform other models in almost all breakdowns, however, its performance comes with the price of computational cost, and face challenges in cyclic graph where it degenerates to enumeration: we further explain in Appendix F.

## 5.3 BENCHMARK RESULTS FOR $k = 2$

As we have explained in Section 4.5, we propose three kinds of metrics, marginal ones, multiply ones, and joint ones, from easy to hard, to evaluate the performance of a model in the scenario of multiple variables. The evaluation result is shown in Table 2. As the effect of the number of constant variables is quite clear, we remove it and add the metrics based on HIT@10 as the new factor.

For the impact regarding the number of existential variables and the topology property of the query graph, we find the result is similar to Table 1, which may be explained by the fact that those models are all initially designed to infer queries with one free variable. For the three metrics we have proposed, we have identified a clear difficulty difference among them though they generally show similar trends. The scores of joint HIT@10 are pretty low, indicating the great hardness of answering queries with multiple variables. Moreover, we have found that FIT falls behind other models in some breakdowns which are mostly cyclic graphs, corroborating our discussion in Section 5.2. We offer more experiment results and further discussion in Appendix H.

## 6 CONCLUSION

In this paper, we make a thorough investigation of the family of $EFO_k$ formulas based on strong theoretical background. We then present a new powerful framework that supports several functionalities essential to CQA task, with this help, we build the $EFO_k$-CQA dataset that greatly extends the previous dataset and benchmark. Our evaluation result brings new empirical findings and reflects the biased selection in the previous dataset impairs the performance of CQA models, emphasizing the contribution of our work.

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

## A  RELATED WORKS

Answering complex queries on knowledge graphs differs from database query answering by being a data-driven task (Wang et al., 2022), where the open-world assumption is addressed by methods that learn from data. Meanwhile, learning-based methods enable faster neural approximate solutions of symbolic query answering problems (Ren et al., 2023).

The prevailing way is query embedding, where the computational results are embedded and computed in the low-dimensional embedding space. Specifically, the query embedding over the set operator trees is the earliest proposed (Hamilton et al., 2018). The supported set operators include projection(Hamilton et al., 2018), intersection (Ren et al., 2020), union and negation (Ren & Leskovec, 2020), and later on be improved by various designs (Zhang et al., 2021; Bai et al., 2022). Such methods assume queries can be converted into the recursive execution of set operations, which imposes additional assumptions on the solvable class of queries (Wang et al., 2021). These assumptions introduce additional limitations of such query embeddings

Recent advancements in query embedding methods adapt query graph representation and graph neural networks, supporting atomics (Liu et al., 2022) and negated atomics (Wang et al., 2023). Query embedding on graphs bypasses the assumptions for queries (Wang et al., 2021). Meanwhile, other search-based inference methods (Arakelyan et al., 2020; Yin et al., 2023) are rooted in fuzzy calculus and not subject to the query assumptions (Wang et al., 2021).

Though many efforts have been made, the datasets of complex query answering are usually subject to the assumptions by set operator query embeddings (Wang et al., 2021). Many other datasets are proposed to enable queries with additional features, see Ren et al. (2023) for a comprehensive survey of datasets. However, only one small dataset proposed by (Yin et al., 2023) introduced queries and

answers beyond such assumptions (Wang et al., 2021). It is questionable that this small dataset is fair enough to justify the advantages claimed in advancement methods (Wang et al., 2023; Yin et al., 2023) that aim at complex query answering. The dataset (Yin et al., 2023) is still far away from the systematical evaluation as Wang et al. (2021) and $\text{EFO}_k$-CQA proposed in this paper fills this gap.

## B  DETAILS OF CONSTRAINT SATISFACTION PROBLEM

In this section, we introduce the constraint satisfaction problem (CSP) again. One instance of CSP $\mathcal{P}$ can be represented by a triple $\mathcal{P} = (X, D, C)$ where $X = (x_1, \cdots, x_n)$ is an $n$-tuple of variables, $D = (D_1, \cdots, D_n)$ is the corresponding $n$-tuple of domains, meaning for each $i$, $x_i \in D_i$. Then, $C = (C_1, \cdots, C_t)$ is $t$-tuple constraint, each constraint $C_i$ is a pair of $(S_i, R_{S_i})$ where $S_i$ is called the scope of the constraint, meaning it is a set of variables $S_i = \{x_{i_j}\}$ and $R_{S_i}$ is the constraint over those variables (Rossi et al., 2006), meaning that $R_{S_i}$ is a subset of the cartesian product of variables in $S_i$.

Then the formulation of existential conjunctive formulas as CSP has already been discussed in Section 2.2. Additionally, for the negation of atomic formula $\neg r(h, t)$, we note the constraint $C$ is also binary with $S_i = \{h, t\}$, $R_{S_i} = \{(h, t) | h, t \in \mathcal{E}, (h, r, t) \notin \mathcal{KG}\}$, this means that $R_{S_i}$ is a very large set, thus the constraint is less "strict" than the positive ones.

## C  PRELIMINARY OF TREE FORM QUERY

We explain the operator tree method, as well as the tree-form queries in this section, which is firstly introduced in Yin et al. (2023). The tree-form queries are defined to be the syntax closure of the operator tree method and are the prevailing query types in the existing datasets (Ren & Leskovec, 2020; Wang et al., 2021), see the definition below:

**Definition 17** (Tree-Form Query). *The set of the Tree-Form queries is the smallest set $\Phi$ such that:*

*(i) If $\phi(y) = r(a, y)$, where $a \in \mathcal{E}$, then $\phi(y) \in \Phi$;*
*(ii) If $\phi(y) \in \Phi$, $\neg\phi(y) \in \Phi$;*
*(iii) If $\phi(y), \psi(y) \in \Phi$, then $(\phi \wedge \psi)(y) \in \Phi$ and $(\phi \vee \psi)(y) \in \Phi$;*
*(iv) If $\phi(y) \in \Phi$ and $y'$ is any variable, then $\psi(y') = \exists y. r(y, y') \wedge \phi(y) \in \Phi$.*

We note that the family of tree-form queries deviates from the targeted $\text{EFO}_1$ query family (Yin et al., 2023). The rationale of the definition is that the previous model relied on the representation of "**operator tree**" which addresses logical queries to simulate logical reasoning as the execution of set operators (Ren & Leskovec, 2020; Zhang et al., 2021; Xu et al., 2022), where each node represents a set of entities corresponding to the answer set of a sub-query (Yin et al., 2023). Then, logical connectives are transformed into operator nodes for set projections(Definition 17 i,iv), complement(Definition 17 ii), intersection, and union(Definition 17 iii) (Wang et al., 2021). Particularly, the set projections are derived from the Skolemization of predicates (Luus et al., 2021). Therefore, the operator tree method that has been adopted in lines of research (Ren & Leskovec, 2020; Zhang et al., 2021; Xu et al., 2022) is just a model that neuralizes these set operations: projection, complement, intersection, and union. These different models basically only differ from each other by their parameterization while having the same expressiveness as characterized by the tree form query.

Specifically, the left side of the Figure 1 shows an example of the operator tree, where "Held" and "Located" are treated as two projections, "N" represents set complement, and "I" represents set intersection. Therefore, the embedding of the root representing the answer set can be computed based on these set operations in a bottom-up manner (Ren & Leskovec, 2020).

Finally, it has been noticed that tree-form query is subject to structural traceability and only has polynomial time combined complexity for inference while the general $\text{EFO}_k$, or even $\text{EFO}_1$ queries, is NP-complete, with detailed proof in Yin et al. (2023). Therefore, this result highlights the importance of investigating the $\text{EFO}_k$ queries as it greatly extends the previous tree-form queries.

## D  CONSTRUCTION OF THE WHOLE $\text{EFO}_k$-CQA DATSET

In this section, we provide details for the construction of the $\text{EFO}_k$-CQA dataset.

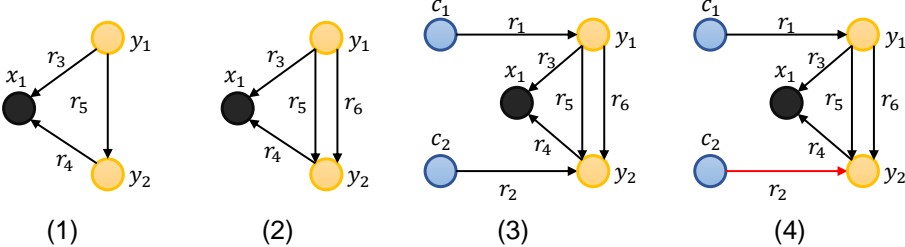

Figure 5: The four steps of enumerating the abstract query graphs. We note that the example and representation follow Figure 3.

### D.1 ENUMERATION OF THE ABSTRACT QUERY GRAPHS

We first give a proposition of the property of abstract query graph:

**Proposition 18.** *For an abstract query graph $\mathcal{G}$, if it conforms Assumption 13 and Assumption 14, then removing all constant entities in $\mathcal{G}$ will lead to only one connected component and no edge is connected between two constant entities.*

*Proof.* We prove this by contradiction. If there is an edge (whether positive or negative) between constant entities, then this edge is redundant, violating Assumption 13. Then, if there is more than one connected component after removing all constant entities in $\mathcal{G}$. Suppose one connected component has no free variable, then this part is a sentence and thus has a certain truth value, whether 0 or 1, which is redundant, violating Assumption 13. Then, we assume every connected component has at least one free variable, we assume there is $m$ connected component and we have:

$$Node(\mathcal{G}) = (\cup_{i=1}^{m} Node(\mathcal{G}_i)) \cup Node(\mathcal{G}_c)$$

where $m > 1$, the $\mathcal{G}_c$ is the set of constant entities and each $\mathcal{G}_i$ is the connected component, we use $Node(\mathcal{G})$ to denote the node set for a graph $\mathcal{G}$. Then this equation describes the partition of the node set of the original $\mathcal{G}$.

Then, we construct $\mathcal{G}_a = G[Node(\mathcal{G}_1) \cup \mathcal{G}_c]$ and $\mathcal{G}_b = G[(\cup_{i=1}^{m} Node(\mathcal{G}_i)) \cup Node(\mathcal{G}_c)]$, where $G$ represents the induced graph. Then we naturally have that $\mathcal{A}[I(\mathcal{G})] = \mathcal{A}[I(\mathcal{G}_a)] \times \mathcal{A}[I(\mathcal{G}_b)]$, where the $\times$ represents the Cartesian product, violating Assumption 14.

$\square$

Additionally, as mentioned in Appendix B, the negative constraint is less "strict", we formally put an additional assumption of the real knowledge graph as the following:

**Assumption 19.** *For any knowledge graph $\mathcal{KG}$, with its entity set $\mathcal{E}$ and relations set $\mathcal{R}$, we assume it is somewhat sparse with regard to each relation, meaning: for any $r \in \mathcal{R}, |\{a \in \mathcal{E}|\exists b.(a, r, b) \in \mathcal{KG} \text{ or } (b, r, a) \in \mathcal{KG}\}| \ll |\mathcal{E}|.$*

Then we develop another proposition for the abstract query graph:

**Proposition 20.** *With the knowledge graph conforming Assumption 19, for any node $u$ in the abstract query graph $\mathcal{G}$, if $u$ is an existential variable or free variable, then it can not only connect with negative edges.*

*Proof.* Suppose $u$ only connects to $m$ negative edge $e_1, \cdots, e_m$. For any grounding $I$, we assume $I(e_i) = r_i \in \mathcal{R}$. For each $r_i$, we construct its endpoint set

$$\text{Endpoint}(r_i) = \{a \in \mathcal{E}|\exists b.(a, r, b) \in \mathcal{KG} \text{ or } (b, r, a) \in \mathcal{KG}\}$$

by the assumption 19, we have $|Endpoint(r_i)| \ll \mathcal{E}|$, then we have:

$$|\cup_{i=1}^{m} \text{Endpoint}(r_i)| \leqslant \Sigma_{i=1}^{m}|\text{Endpoint}(r_i)| \ll |\mathcal{E}|$$

since $m$ is small due to the size of the abstract query graph. Then we have two situations about the type of node $u$:

**1.If node $u$ is an existential variable.**

Then we construct a subgraph $\mathcal{G}_s$ be the induced subgraph of $Node(\mathcal{G}) - u$, then for any possible grounding $I$, we prove that $\mathcal{A}[I(\mathcal{G}_s)] = \mathcal{A}[I(\mathcal{G})]$, the right is clearly a subset of the left due to it contains more constraints, then we show every answer of the left is also an answer on the right, we merely need to give an appropriate candidate in the entity set for node $v$, and in fact, we choose any entity in the set $\mathcal{E} - \cup_{i=1}^{m}\text{Endpoint}(r_i)$ since it suffices to satisfies all constraints of node $u$, and we have proved that $|\mathcal{E} - \cup_{i=1}^{m}\text{Endpoint}(r_i)| > 0$.

This violates the Assumption 13.

**2.If node $u$ is a free variable.**

Similarly, any entity in the set $\mathcal{E} - \cup_{i=1}^{m}\text{Endpoint}(r_i)$ will be an answer for the node $u$, thus violating the Assumption 16.

$\square$

We note the proposition 20 extends the previous requirement about negative queries, which is firstly proposed in Ren & Leskovec (2020) and inherited and named as "bounded negation" in Wang et al. (2021), the "bounded negation" requires the negation operator should be followed by the intersection operator in the operator tree. Obviously, the abstract query graph that conforms to "bounded negation" will also conform to the requirement in Proposition 20. A vivid example is offered in Figure 2.

Finally, we make the assumption of the distance to the free variable of the query graph:

**Assumption 21.** *There is a constant $d$, such that for every node $u$ in the abstract query graph $\mathcal{G}$, it can find a free variable in its $d$-hop neighbor.*

We have this assumption to exclude the extremely long-path queries.

Equipped with the propositions and assumptions above, we explore the combinatorial space of the abstract query graph given certain hyperparameters, including: the max number of free variables, max number of existential variables, max number of constant entities, max number of all nodes, max number of all edges, max number of edges surpassing the number of nodes, max number of negative edge, max distance to the free variable. In practice, these numbers are set to be: 2, 2, 3, 6, 6, 0, 1, 3. We note that the max number of edges surpassing the number of nodes is set to 0, which means that the query graph can at most have one more edge than a simple tree, thus, we exclude those query graphs that are both cyclic graphs and multigraphs, making our categorization and discussion in the experiments in Section 5.2 and Section 5.3 much more straightforward and clear.

Then, we create the abstract query graph by the following steps, which is a graph with three types of nodes and two kinds of edges:

1. First, create a simple connected graph $\mathcal{G}_1$ with two types of nodes, the existential variable and the free variable, and one type of edge, the positive edge.
2. We add additional edges to the simple graph $\mathcal{G}_1$ and make it a multigraph $\mathcal{G}_2$.
3. Then, the constant variable is added to the graph $\mathcal{G}_2$, In this step, we make sure not too long existential leaves. The result is graph $\mathcal{G}_3$.
4. Finally, random edges in $\mathcal{G}_3$ are replaced by the negation edge, and we get the final abstract query graph $\mathcal{G}_4$.

In this way, all possible query graphs within a certain combinatorial space are enumerated, and finally, we filter duplicated graphs with the help of the graph isomorphism algorithm. We give an example to illustrate the four-step construction of an abstract query graph in Figure 5.

### D.2 GROUND ABSTRACT QUERY GRAPH WITH MEANINGFUL NEGATION

To fulfill the Assumption 15 as discussed in Section 4.2, for an abstract query graph $\mathcal{G} = (V, E, f, g)$, we have two steps: (1). Sample grounding for the positive subgraph $\mathcal{G}_p$ and compute its answer (2).

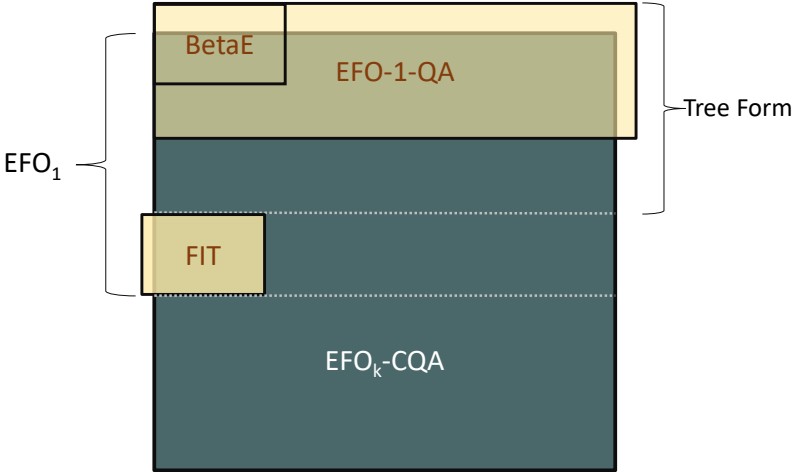

Figure 6: Illustration of the comparison between the $EFO_k$-CQA dataset (navy blue box) and the previous dataset (three yellow boxes), where the BetaE and EFO-1-QA aim to investigate the tree form query, explained in Appendix C, while the FIT dataset aims to investigate $EFO_1$ query that is not tree form. FIT is not a subset of $EFO_k$-CQA because its "3pm" query is not included in $EFO_k$-CQA.

Ground the $\mathcal{G}_n$ to decrease the answer got in the first step. Then we define positive subgraph $\mathcal{G}_p$ to be defined as such, its edge set $E' = \{e \in E | g(e) = positive\}$, its node set $V' = \{u | u \in V, \exists e \in E'$ and $e$ connects to $u\}$. Then $\mathcal{G}_p = (V', E', f, g)$. We note that because of Proposition 20, if a node $u \in V - V'$, then we know node $u$ must be a constant entity.

Then we sample the grounding for the positive subgraph $\mathcal{G}_p$, we also compute the CSP answer $\overline{\mathcal{A}}_p$ for this subgraph.

Then we ground what is left in the positive subgraph, we split each negative edge in $E - E'$ into two categories:

**1. This edge $e$ connects two nodes $u, v$, and $u, v \in V'$.**

In this case, we sample the relation $r$ to be the grounding of $e$ such that it negates some of the answers in $\overline{\mathcal{A}}_p$.

**2. This edge $e$ connects two nodes $u, v$, where $u \in V'$, while $v \notin V'$.**

In this case, we sample the relation $r$ for $e$ and entity $a$ for $v$ such that they negate some answer in $\overline{\mathcal{A}}_p$, we note we only need to consider the possible candidates for node $u$ and it is quite efficient.

We note that there is no possibility that neither of the endpoints is in $V'$ because as we have discussed above, this means that both nodes are constant entities, but in Proposition 18 we have asserted that no edge is connected between two entities.

### D.3 THE COMPARISON TO PREVIOUS BENCHMARK

To give an intuitive comparison of our $EFO_k$-CQA dataset against those previous datasets and benchmark, including the BetaE dataset (Ren & Leskovec, 2020), the EFO-1-QA benchmark (Wang et al., 2021) that extends BetaE dataset, and the FIT dataset (Yin et al., 2023) that explores 10 more new query types, we offer a new figure in Figure 6.

It can be clearly observed that EFO-1-QA covers the BetaE dataset and has provided a quite systematic investigation in tree form query, while FIT deviates from them and studies ten new query types that are in $EFO_1$ but not tree form.

As discussed in Section 3, the scope of the formula investigated in our $EFO_k$-CQA dataset surpasses the previous EFO-1-QA benchmark and FIT dataset because of three reasons: (1). We include

Table 3: The number of abstract query graphs with one free variable. We denote $e$ as the number of existential variables and $c$ as the number of constant entities. SDAG represents the Simple Directed Acyclic Graph, Multi for multigraph, and Cyclic for the cyclic graph. Sum.$(c)$ and Sum.$(e)$ is the total number of queries with the number of constant entities / existential variables fixed.

| $c$ \ $e$ | 0 | 1 | | 2 | | | Sum.$(c)$ | Sum. |
|---|---|---|---|---|---|---|---|---|
| | SDAG | SDAG | Multi | SDAG | Multi | Cyclic | | |
| 1 | 1 | 2 | 4 | 4 | 16 | 4 | 31 | |
| 2 | 2 | 6 | 6 | 20 | 40 | 8 | 82 | 251 |
| 3 | 2 | 8 | 8 | 36 | 72 | 12 | 138 | |
| Sum.$(e)$ | 5 | 16 | 18 | 60 | 128 | 24 | | |

the $\text{EFO}_k$ formula with multiple free variables that has never been investigated(the bottom part of navy blue box in Figure 6); (2). We systematically investigate those $\text{EFO}_1$ queries that are not tree form while the previous FIT dataset only discusses ten hand-crafted query types (the navy blue part between two white lines in Figure 6); (3) Our assumption is more systematic than previous ones as shown by the example in Figure 2(the top navy blue part above two white lines in Figure 6). Though we only contain 741 query types while the EFO-1-QA benchmark contains 301 query types, we list reasons for the number of query types is not significantly larger than the previous benchmark: (1). EFO-1-QA benchmark relies on the operator tree that contains union, which represents the logic conjunction($\lor$), however, we only discuss the conjunctive queries because we always utilize the DNF of a query. We notice that there are only 129 query types in EFO-1-QA without the union, significantly smaller than the $\text{EFO}_k$-CQA dataset. (2). In the construction of $\text{EFO}_k$-CQA dataset, we restrict the query graph to have at most one negative edge to avoid the total number of query types growing quadratically, while in EFO-1-QA benchmark, their restrictions are different than ours and it contains queries that have two negative atomic formulas as indicated by the right part of yellow box is not contained in the navy blue box.

## D.4 $\text{EFO}_k$-CQA STATISTICS

The statistics of our $\text{EFO}_k$-CQA dataset are shown in Table 3 and Table 4, they show the statistics of our abstract query graph by their topology property, the statistics are split into the situation that the number of free variable $k = 1$ and the number of free variable $k = 2$, correspondingly. We note abstract query graphs with seven nodes have been excluded as the setting of hyperparameters discussed in Appendix D.1, we make these restrictions to control the quadratic growth in the number of abstract query graphs.

Finally, in FB15k-237, we sample 1000 queries for an abstract query graph without negation, 500 queries for an abstract query graph with negation; in FB15k, we sample 800 queries for an abstract query graph without negation, 400 queries for an abstract query graph with negation; in NELL, we sample 400 queries for an abstract query graph without negation, 100 queries for an abstract query graph with negation. As we have discussed in Appendix D.2, sample negative query is computationally costly, thus we sample less of them.

Moreover, we provide our $\text{EFO}_k$-CQA dataset an inductive version, with the same query types as the transductive version, while the number of queries per query type is set to 400 for positive ones and 100 for negative ones. The inductive ratio is set to 175%, following the setting in Galkin et al. (2022).

## E EVALUATION DETAILS

We explain the evaluation protocol in detail for Section 4.5.

Firstly, we explain the computation of common metrics, including Mean Reciprocal Rank(MRR) and HIT@K, given the full answer $\mathcal{A}$ in the whole knowledge graph and the observed answer $\mathcal{A}_o$ in the observed knowledge graph, we focus on the hard answer $\mathcal{A}_h$ as it requires more than memorizing the observed knowledge graph and serves as the indicator of the capability of reasoning.

---

**Algorithm 1** Embedding computation on the query graph.

---

**Require:** The query graph $G$.

  Compute the ordering of the nodes as explained in Algorithm 2.

  Create a dictionary $E$ to store the embedding for each node in the query graph

  **for** $i \leftarrow 1$ to $n$ **do**

    **if** node $u_i$ is a constant entity **then**

      The embedding of $u_i$, $E[i]$ is gotten from the entity embedding

    **else**

      Then we know node $u_i$ is either free variable or existential variable

      Compute the set of nodes $\{u_{i_j}\}_{j=1}^t$ that are previous to $i$ and adjacency to node $u_i$.

      Create a list to store projection embedding $L$.

      **for** $j \leftarrow 1$ to $t$ **do**

        Find the relation $r$ between node $u_i$ and $u_{i_j}$, get the embedding of node $u_{i_j}$ as $E[i_j]$.

        **if** $E[i_j]$ is not None **then**

          **if** The edge between $u_i$ and $u_{i_J}$ is positive **then**

            Compute the embedding of projection($E[i_j], r$), add it to the list $L$.

          **else**

            Compute the embedding of the negation of the projection($E[i_j], r$), add it to the list $L$.

          **end if**

        **end if**

      **end for**

      **if** The list $L$ has no element **then**

        $E[i]$ is set to none.

      **else if** The list $L$ has one element **then**

        $E[i] = L[0]$

      **else**

        Compute the embedding as the intersection of the embedding in the list $L$, and set $E[i]$ as the outcome.

      **end if**

    **end if**

  **end for**

  **return** The embedding dictionary $E$ for each node in the query graph.

---

---

**Algorithm 2** Node ordering on the abstract query graph.

---

**Require:** The abstract query graph $\mathcal{G} = (V, E, f, g)$, $V$ consists $m$ nodes, $u_1, \cdots, u_m$.

Creates an empty list $L$ to store the ordering of the node.

Creates another two set $S_1$ and $S_2$ to store the nodes that are to be explored next.

**for** $i \leftarrow 1$ to $m$ **do**

    **if** The type of node $f(u_i)$ is constant entity **then**

        list $L$ append the node $u_i$

        **for** Node $u_j$ that connects to $u_i$ **do**

            **if** $f(u_j)$ is existential variable **then**

                $u_j$ is added to set $S_1$

            **else**

                $u_j$ is added to set $S_2$

            **end if**

        **end for**

    **end if**

    **while** Not all node is included in $L$ **do**

        **if** Set $S_1$ is not empty **then**

            We sort the set $S_1$ by the sum of their distance to every free variable in $\mathcal{G}$, choose the most remote one, and if there is a tie, randomly choose one node, $u_i$ to be the next to explore. We remove $u_i$ from set $S_1$.

        **else**

            In this case, we know set $S_2$ is not empty because of the connectivity of $\mathcal{G}$. We randomly choose a node $u_i \in S_2$ to be the next node to explore. We remove $u_i$ from set $S_2$.

        **end if**

        **for** Node $u_j$ that connects to $u_i$ **do**

            **if** $f(u_j)$ is existential variable **then**

                $u_j$ is added to set $S_1$

            **else**

                $u_j$ is added to set $S_2$

            **end if**

        **end for**

        List $L$ append the node $u_i$

    **end while**

**end for**

**return** The list $L$ as the ordering of nodes in the whole abstract query graph $\mathcal{G}$

---

Table 4: The number of abstract query graphs with two free variables. The notation of $e$, $c$ SDAG, Multi, and Cyclic are the same as Table 3. And "-" means that this type of abstract query graph is not included.

| $c$ $\diagdown$ $e$ | $e = 0$ | | $e = 1$ | | | $e = 2$ | | | AVG. |
|---|---|---|---|---|---|---|---|---|---|
| | SDAG | Multi | SDAG | Multi | Cyclic | SDAG | Multi | Cyclic | |
| $c = 1$ | 1 | 2 | 7 | 18 | 4 | 6 | 32 | 26 | 96 |
| $c = 2$ | 4 | 4 | 20 | 36 | 8 | 38 | 108 | 64 | 282 |
| $c = 3$ | 4 | 4 | 32 | 60 | 12 | - | - | - | 112 |

Specifically, we rank each hard answer $a \in \mathcal{A}_h$ against all non-answers $\mathcal{E} - \mathcal{A} - \mathcal{A}_o$, the reason is that we need to neglect other answers so that answers do not interfere with each other, finally, we get the ranking for $a$ as $r$. Then its MRR is $1/r$, and its HIT@k is $\mathbf{1}_{r \leqslant k}$, thus, the score of a query is the mean of the scores of every its hard answer. We usually compute the score for a query type (which corresponds to an abstract query graph) as the mean score of every query within this type.

As the marginal score and the multiply score have already been explained in Section 4.5, we only mention one point that it is possible that every free variable does not have marginal hard answer. Assume that for a query with two free variables, its answer set $\mathcal{A} = \{(a_1, a_2), (a_1, a_3), (a_4, a_2)\}$ and its observed answer set $\mathcal{A}_o = \{(a_1, a_3), (a_4, a_2)\}$. In this case, $a_1$ is not the marginal hard answer for the first free variable and $a_2$ is not the marginal hard answer for the second free variable, in general, no free variable has its own marginal hard answer.

Then we only discuss the joint metric, specifically, we only explain how to estimate the joint ranking by the individual ranking of each free variable. For each possible $k$-tuple $(a_1, \cdots, a_k)$, if $a_i$ is ranked as $r_i$ among the **whole** entity set $\mathcal{E}$, we compute the score of this tuple as $\Sigma_{i=1}^{k} r_i$, then we sort the whole $\mathcal{E}^k$ $k$-tuple by their score, for the situation of a tie, we just use the lexicographical order. After the whole joint ranking is got, we use the standard evaluation protocol that ranks each hard answer against all non-answers. It can be confirmed that this estimation method admits a closed-form solution for the sorting in $\mathcal{E}^k$ space, thus the computation cost is affordable.

We just give the closed-form solution when there are two free variables:

for the tuple $(r_1, r_2)$, the possible combinations that sum less than $r_1 + r_2$ is $\binom{r_1 + r_2 - 1}{2}$, then, there is $r_1 - 1$ tuple that ranks before $(r_1, r_2)$ because of lexicographical order, thus, the final ranking for the tuple $(r_1, r_2)$ is just $\binom{r_1 + r_2 - 1}{2} + r_1$ that can be computed efficiently.

## F  IMPLEMENTATION DETAILS OF CQA MODELS

In this section, we provide implementation details of CQA models that have been evaluated in our paper. For query embedding methods that rely on the operator tree, including BetaE (Ren & Leskovec, 2020), LogicE (Luus et al., 2021), and ConE (Zhang et al., 2021), we compute the ordering of nodes in the query graph in Algorithm 2, then we compute the embedding for each node in the query graph Algorithm 1, the final embedding of every free node are gotten to be the predicted answer. Especially, the node ordering we got in Algorithm 2 coincides with the natural topology ordering induced by the directed acyclic operator tree, so we can compute the embedding in the same order as the original implementation. Then, in Algorithm 1, we implement each set operation in the operator tree, including intersection, negation, and set projection. By the merit of the Disjunctive Normal Form (DNF), the union is tackled in the final step. Thus, our implementation can coincide with the original implementation in the original dataset (Ren & Leskovec, 2020).

For CQD (Arakelyan et al., 2020) and LMPNN (Wang et al., 2023), their original implementation does not require the operator tree, so we just use their original implementation. Specifically, in a query graph with multiple free variables, for CQD we predict the answer for each free variable individually as taking others free variables as existential variables, for LMPNN, we just got all embedding of nodes that represent free variables.

For FIT (Yin et al., 2023), though it is proposed to solve $EFO_1$ queries, it is computationally costly: it has a complexity of $O(\mathcal{E}^2)$ in the acyclic graphs and is even not polynomial in the cyclic graphs,

the reason is that FIT degrades to enumeration to deal with cyclic graph. In our implementation, we further restrict FIT to at most enumerate 10 possible candidates for each node in the query graph, this practice has allowed FIT to be implemented in the dataset FB15k-237 (Toutanova & Chen, 2015). However, it cost 20 hours to evaluate FIT on our $EFO_k$-CQA dataset while other models only need no more than two hours. Moreover, for larger knowledge graph, including NELL (Carlson et al., 2010) and FB15k (Bordes et al., 2013), we have also encountered an out-of-memory error in a Tesla V100 GPU with 32G memory when implementing FIT, thus, we omit its result in these two knowledge graphs.

## G  EXTENSION TO MORE COMPLEX QUERY ANSWERING

In this section, we discuss possible further development in the task of complex query answering and how our work, especially our framework proposed in Section 4 can help with future development. We list some new features that may be of interest and show the maximum versatility our framework can reach. Our analysis and characterization of future queries inherit the outlook in Wang et al. (2022) and also is based on the current development.

**Inductive Reasoning** Inductive reasoning is a new trend in the field of complex query answering. Some entities (Galkin et al., 2022) or even relations (Huang et al., 2022) are not seen in the training period, namely they can not be found by the observed knowledge graph $\mathcal{G}_o$ therefore, the inductive generalization is essential for the model to infer answers. We note that our framework is powerful enough to sample inductive queries with the observed knowledge graph $\mathcal{G}_o$ given. Therefore, the functionality of sampling inductive query is easily contained and implemented in our framework, see `https://anonymous.4open.science/r/EFOK-CQA/README.md`. We note there we have already provided our $EFO_k$-CQA dataset in this setting as discussed in Appendix D.4.

**N-ary relation** N-ary relation is a relation that has $n > 2$ corresponding entities, therefore, the factual information in the knowledge graph is not a triple but a $(n + 1)$-tuple. Moreover, the query graph is also a hypergraph, making the corresponding CSP problem even harder. This is a newly introduced topic (Luo et al., 2022; Alivanistos et al., 2022) in complex query answering, which our framework has limitations in representing.

**Knowledge graph with attribute** Currently, there has been some research that has taken the additional attribute of the knowledge graph into account. Typical attributes include entity types (Hu et al., 2022), numerical literals (Bai et al., 2023),triple timestamps (Jia et al., 2021; Saxena et al., 2021), and triple probabilities (Carlson et al., 2010). We note that attributes expand the entity set $\mathcal{E}$ from all entities to entities with attribute values, it is also possible that the relation set $\mathcal{R}$ is also extended to contain corresponding relations, like "greater", "less" when dealing with numerical literals. Then, our framework can represent queries on such extended knowledge graphs like in Bai et al. (2023), where no function like "plus", or "minus" is considered and the predicates are also binary.

Overall, our framework can be applied to some avant-garde problem settings given certain properties, thus those functionalities proposed in Section 4 can be useful. We hope our discussion helps with the future development of complex query answering.

## H  ADDITIONAL EXPERIMENT RESULT AND ANALYSIS

In this section, we offer another experiment result not available to be shown in the main paper. For the purpose of supplementation, we select some representative experiment result as the experiment result is extremely complex to be categorized and be shown. we present the further benchmark result of the following: the analysis of benchmark result in detail, more than just the averaged score in Table 1 and Table 2, which is provided in Appendix H.1; result of different knowledge graphs, including NELL and FB15k, which is provided in Appendix H.2 and H.3, the situation of more constant entities since we only discuss when there are two constant entities in Table 2, the result is provided in Appendix H.4, and finally, all queries(including the queries without marginal hard answers), in Appendix H.5.

We note that we have explained in Section 4.5 and Appendix E that for a query with multiple free variables, some or all of the free variables may not have their marginal hard answer and thus the marginal metric can not be computed. Therefore, in the result shown in Table 2 in Section 5.3, we

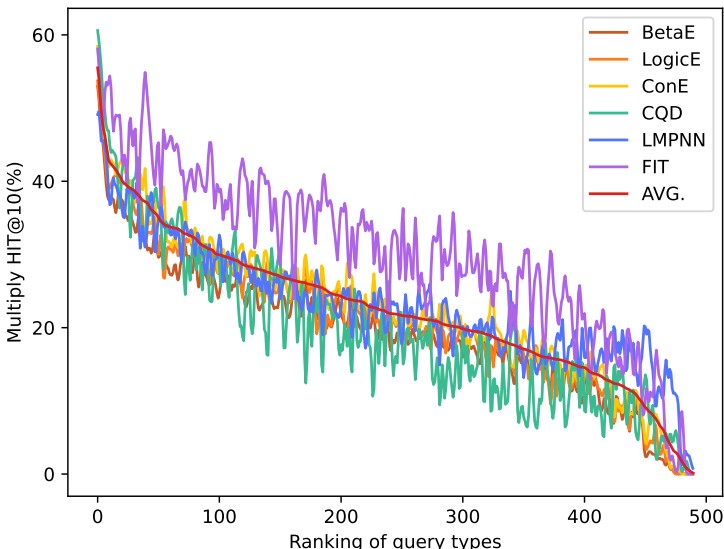

Figure 7: Relative performance of the six representative CQA models in referring queries with two free variables, the ranking of query types is determined by the average Multiply HIT@10 score. A Gaussian filter with sigma=1 is added to smooth the curve.

only conduct evaluation on those queries that both of their free variables have marginal hard answers, and we offer the benchmark result of all queries in Appendix H.5 where only two kinds of metrics are available.

## H.1    FURTHER RESULT AND ANALYSIS OF THE EXPERIMENT IN MAIN PAPER

To supplement the experiment result already shown in Section 5.2 and Section 5.3, we have included more benchmark results in this section. Though the averaged score is a broadly-used statistic to benchmark the model performance on our $EFO_k$ queries, this is not enough and we have offered much more detail in this section.

**Whole combinatorial space helps to develop trustworthy machine learning models.** Firstly, we show more detailed benchmark results of the relative performance between our selected six CQA models, the result is shown in Table 4. Specifically, we plot two boxes, the black one, including the most difficult query types, and the red box, including the easiest query types. In the easiest part, we find that even the worst model and the best model have pretty similar performance despite that they may differ greatly in other query types. The performance in the most difficult query types is more important when the users are risk-sensitive and desire a trustworthy machine-learning model that does not crash in extreme cases (Varshney, 2019) and we highlight it in the black box. In the black box, we note that CQD (Arakelyan et al., 2020), though designed in a rather general form, is pretty unstable when comes to empirical evaluation, as it has a clear downward curve and deviates from other model's performance enormously in the most difficult query types. Therefore, though its performance is better than LMPNN and comparable to BetaE on average as reported in Table 1, its unsteady performance suggests its inherent weakness. On the other hand, ConE (Zhang et al., 2021) is much more steady and outperforms BetaE and LogicE consistently. We also show the result when there are two free variables in Figure 7, where the model performance is much less steady but the trend is similar to the $EFO_1$ case in general.

**Empirical hardness of query types and incomplete discussion of the previous dataset.** Moreover, we also discuss the empirical hardness of query types themselves and compare different datasets accordingly in Figure 8. We find the standard deviation of the six representative CQA models increases in the most difficult part and decreases in the easiest part, corroborating our discussion

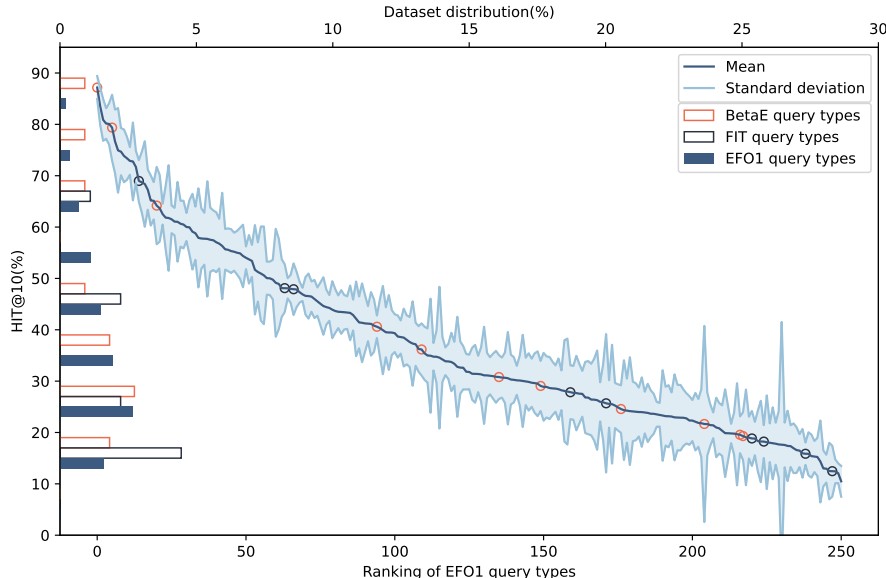

Figure 8: Query type distribution in three different datasets, BetaE one, FIT one, and the EFO$_1$ part in our EFO$_k$-CQA dataset. The left part shows the histogram that represents the probability density function of each dataset. The ranking of query types is also determined by the mean HIT@10 score as in Figure 4, with the standard deviation of the performance of the six CQA models shown as the light blue error bar.

in the first paragraph. We also highlight those query types that have already been investigated in BetaE dataset (Ren & Leskovec, 2020) and FIT dataset (Yin et al., 2023). We intuitively find that the BetaE dataset does not include very challenging query types while the FIT dataset mainly focuses on them. This can be explained by the fact that nine out of ten most challenging query types correspond to multigraph, which the BetaE dataset totally ignores while the FIT dataset highlights it as a key feature. To give a quantitative analysis of whether their hand-crafted query types are sampled from the whole combinatorial space, we have adopted the Kolmogorov–Smirnov test to test the distribution discrepancy between their distribution and the query type distribution in EFO$_k$-CQA since EFO$_k$-CQA enumerates all possible query types in the given combinatorial space and is thus unbiased. We find that the BetaE dataset is indeed generally easier and its p-value is 0.78, meaning that it has a 78 percent possibility to be unbiased, while the FIT dataset is significantly harder and its p-value is 0.27. Therefore, there is no significant statistical evidence to prove they are sampled from the whole combinatorial space unbiasedly.

## H.2 FURTHER BENCHMARK RESULT OF $k$=1

Firstly, we present the benchmark result when there is only one free variable, since the result in FB15k-237 is provided in Table 1, we provide the result for other standard knowledge graphs, FB15k and NELL, their result is shown in Table 6 and Table 7, correspondingly. We note that FIT is out of memory with the two large graphs FB15k and NELL as explained in Appendix F and we do not include its result. As FB15k and NELL are both reported to be easier than FB15k-237, the models have better performance. The trend and analysis are generally similar to our discussion in Section 5.2 with some minor, unimportant changes that LogicE (Luus et al., 2021) has outperformed ConE (Zhang et al., 2021) in the knowledge graph NELL, indicating one model may not perform identically well in all knowledge graphs.

Table 5: MRR scores(%) for inferring queries with one free variable on FB15k-237. We denote $e$ as the number of existential variables and $c$ as the number of constant entities. SDAG represents the Simple Directed Acyclic Graph, Multi for multigraph, and Cyclic for the cyclic graph. AVG.($c$) and AVG.($e$) is the average score of queries with the number of constant entities / existential variables fixed.

| Model | $c$ \ $e$ | 0 | 1 | | 2 | | | AVG.($c$) | AVG. |
|---|---|---|---|---|---|---|---|---|---|
| | | SDAG | SDAG | Multi | SDAG | Multi | Cyclic | | |
| BetaE | 1 | 16.2 | 17.9 | 10.9 | 10.6 | 8.5 | 16.5 | 11.1 | |
| | 2 | 35.6 | 20.2 | 19.1 | 15.7 | 15.7 | 27.1 | 17.8 | 20.7 |
| | 3 | 53.3 | 32.4 | 33.1 | 21.7 | 21.6 | 37.4 | 24.8 | |
| | AVG.($e$) | 37.4 | 25.7 | 23.5 | 18.8 | 18.1 | 30.5 | | |
| LogicE | 1 | 17.4 | 19.0 | 11.5 | 11.0 | 8.5 | 16.8 | 11.5 | |
| | 2 | 36.7 | 21.2 | 19.8 | 16.5 | 16.1 | 27.3 | 18.4 | 21.3 |
| | 3 | 55.5 | 34.6 | 34.5 | 22.3 | 22.0 | 37.5 | 25.4 | |
| | AVG.($e$) | 38.9 | 27.3 | 24.5 | 19.4 | 18.5 | 30.6 | | |
| ConE | 1 | 18.6 | 19.9 | 11.8 | 11.4 | 9.3 | 18.7 | 12.3 | |
| | 2 | 39.1 | 22.4 | 20.8 | 18.1 | 17.6 | 30.7 | 20.1 | 23.1 |
| | 3 | 58.8 | 36.4 | 37.0 | 24.6 | 23.8 | 41.7 | 27.6 | |
| | AVG.($e$) | 41.4 | 28.7 | 26.0 | 21.3 | 20.1 | 34.2 | | |
| CQD | 1 | **22.2** | 19.5 | 9.0 | 9.2 | 6.4 | 15.6 | 10.0 | |
| | 2 | 35.3 | 20.1 | 19.1 | 16.4 | 16.2 | 27.6 | 18.4 | 21.9 |
| | 3 | 40.3 | 32.9 | 34.3 | 24.4 | 24.0 | 40.2 | 26.8 | |
| | AVG.($e$) | 33.9 | 26.2 | 23.7 | 20.5 | 19.4 | 31.9 | | |
| LMPNN | 1 | 20.5 | 21.4 | 11.2 | 11.6 | 8.7 | 17.0 | 11.9 | |
| | 2 | 42.0 | 22.6 | 18.5 | 16.5 | 14.9 | 26.5 | 17.9 | 20.5 |
| | 3 | 62.3 | 35.9 | 31.6 | 22.1 | 19.8 | 35.5 | 24.0 | |
| | AVG.($e$) | 44.2 | 28.8 | 22.7 | 19.4 | 16.9 | 29.4 | | |
| FIT | 1 | **22.2** | **25.0** | **17.4** | **13.9** | **11.7** | **23.3** | **15.6** | |
| | 2 | **45.3** | **29.6** | **28.5** | **23.8** | **24.3** | **35.5** | **26.5** | **30.3** |
| | 3 | **64.5** | **44.8** | **45.4** | **33.3** | **33.5** | **44.4** | **36.2** | |
| | AVG.($e$) | **46.7** | **36.2** | **33.6** | **28.6** | **27.9** | **37.9** | | |

## H.3 FURTHER BENCHMARK RESULT FOR $k$=2 IN MORE KNOWLEDGE GRAPHS

Then, similar to Section 5.3, we provide the result for other standard knowledge graphs, FB15k and NELL, when the number of constant entities is fixed to two, their result is shown in Table 8 and Table 9, correspondingly.

We note that though in some breakdowns, the marginal score is over 90 percent, almost close to 100 percent, the joint score is pretty slow, which further corroborates our findings that joint metric is significantly harder and more challenging in Section 5.3.

## H.4 FURTHER BENCHMARK RESULT FOR $k$=2 WITH MORE CONSTANT NUMBERS.

As the experiment in Section 5.3 only contains the situation where the number of constant entity is fixed as one, we offer the further experiment result in Table 10.

The result shows that models perform worse with fewer constant variables when compares to the result in Table 2, this observation is the same as the previous result with one free variable that has been discussed in Section 5.2.

Table 6: MRR scores(%) for inferring queries with one free variable on FB15k. The notation of $e$, $c$, SDAG, Multi, Cyclic, AVG.($c$) and AVG.($e$) are the same as Table 1.

| Model | $c$ \ $e$ | 0 SDAG | 1 SDAG | 1 Multi | 2 SDAG | 2 Multi | 2 Cyclic | AVG.($c$) | AVG. |
|---|---|---|---|---|---|---|---|---|---|
| BetaE | 1 | 38.6 | 30.4 | 29.2 | 21.7 | 21.7 | 24.1 | 24.3 | |
| | 2 | 49.7 | 34.0 | 37.2 | 28.3 | 29.2 | 35.5 | 31.0 | 34.0 |
| | 3 | 63.5 | 46.4 | 48.6 | 33.9 | 36.1 | 45.8 | 38.1 | |
| | AVG.($e$) | 63.5 | 46.4 | 48.6 | 33.9 | 36.1 | 45.8 | 38.1 | |
| LogicE | 1 | 46.0 | 33.8 | 32.1 | 23.3 | 22.8 | 25.6 | 26.2 | |
| | 2 | 51.2 | 35.9 | 39.0 | 30.6 | 30.5 | 36.9 | 32.7 | 35.6 |
| | 3 | 64.5 | 48.6 | 49.8 | 35.4 | 37.5 | 47.7 | 39.6 | |
| | AVG.($e$) | 54.9 | 41.7 | 42.3 | 32.8 | 33.4 | 40.4 | | |
| ConE | 1 | 52.5 | 35.8 | 34.9 | 25.9 | 25.9 | 29.5 | 29.3 | |
| | 2 | 57.0 | 40.0 | 43.4 | 33.2 | 34.2 | 40.8 | 36.3 | 39.5 |
| | 3 | 70.6 | 53.1 | 55.3 | 39.3 | 41.8 | 52.5 | 43.9 | |
| | AVG.($e$) | 61.0 | 45.6 | 46.8 | 36.1 | 37.4 | 44.8 | | |
| CQD | 1 | 74.6 | 36.1 | 32.7 | 17.6 | 16.7 | 25.4 | 23.7 | |
| | 2 | 52.2 | 35.2 | 40.9 | 29.2 | 31.5 | 39.2 | 33.2 | 37.2 |
| | 3 | 53.3 | 32.4 | 33.1 | 21.7 | 21.6 | 37.4 | 24.8 | |
| | AVG.($e$) | 59.4 | 41.5 | 44.6 | 33.3 | 35.3 | 43.3 | | |
| LMPNN | 1 | 63.7 | 39.9 | 35.3 | 28.7 | 26.4 | 28.7 | 30.7 | |
| | 2 | 65.0 | 41.9 | 38.8 | 34.4 | 31.7 | 38.4 | 35.1 | 37.7 |
| | 3 | 79.8 | 54.0 | 49.5 | 38.9 | 37.1 | 48.0 | 40.8 | |
| | AVG.($e$) | 70.2 | 47.4 | 42.8 | 36.6 | 34.1 | 41.6 | | |

## H.5 FURTHER BENCHMARK RESULT FOR $k$=2 INCLUDING ALL QUERIES

Finally, as we have explained in Section 4.5 and Appendix E, there are some valid $\text{EFO}_k$ queries without marginal hard answers when $k > 1$. Thus, there is no way to calculate the marginal scores, all our previous experiments are therefore only conducted on those queries that all their free variables have marginal hard answers. In this section, we only present the result of the Multiply and Joint score, as they can be computed for any valid $\text{EFO}_k$ queries, and therefore this experiment is conducted on the whole $\text{EFO}_k$-CQA dataset.

We follow the practice in Section 5.3 that fixed the number of constant entities as two, as the impact of constant entities is pretty clear, which has been further corroborated in Appendix H.4. The experiments are conducted on all three knowledge graphs, FB15k-237, FB15k, and NELL, the result is shown in Table 11, Table 12, and Table 13, correspondingly.

Interestingly, comparing the result in Table 2 and Table 11, the multiple scores actually increase through the joint scores are similar. This may be explained by the fact that if one free variable has no marginal hard answer, then it can be easily predicted, leading to a better performance for the whole query.

Table 7: MRR scores(%) for inferring queries with one free variable on NELL. The notation of $e$, $c$, SDAG, Multi, Cyclic, AVG.($c$) and AVG.($e$) are the same as Table 1.

| Model | $e$ \\ $c$ | 0 SDAG | 1 SDAG | 1 Multi | 2 SDAG | 2 Multi | 2 Cyclic | AVG.($c$) | AVG. |
|-------|------------|--------|--------|---------|--------|---------|----------|-----------|------|
| BetaE | 1 | 13.9 | 26.4 | 35.0 | 8.6 | 14.9 | 19.1 | 17.5 | |
|       | 2 | 58.8 | 31.5 | 43.8 | 22.4 | 30.6 | 34.7 | 30.7 | 33.6 |
|       | 3 | 78.8 | 48.6 | 58.3 | 29.6 | 39.0 | 47.0 | 39.5 | |
|       | AVG.($e$) | 53.1 | 38.5 | 48.3 | 25.2 | 33.3 | 38.2 | | |
| LogicE | 1 | 18.3 | 29.2 | 39.6 | 12.1 | 19.0 | 20.4 | 21.1 | |
|        | 2 | 63.5 | 34.4 | 47.3 | 26.4 | 34.0 | 37.6 | 34.2 | 36.9 |
|        | 3 | 79.6 | 51.2 | 59.3 | 33.1 | 42.2 | 50.1 | 42.6 | |
|        | AVG.($e$) | 56.3 | 41.3 | 50.9 | 28.8 | 36.7 | 41.0 | | |
| ConE | 1 | 16.7 | 26.9 | 36.6 | 11.1 | 16.9 | 22.3 | 19.6 | |
|      | 2 | 60.5 | 33.6 | 46.6 | 25.3 | 33.1 | 40.1 | 33.6 | 36.6 |
|      | 3 | 79.9 | 50.6 | 59.2 | 33.2 | 42.2 | 52.6 | 42.8 | |
|      | AVG.($e$) | 54.9 | 40.3 | 50.0 | 28.4 | 36.2 | 43.4 | | |
| CQD | 1 | 22.3 | 30.6 | 37.3 | 13.3 | 17.9 | 20.7 | 20.9 | |
|     | 2 | 59.8 | 34.0 | 45.2 | 28.8 | 35.4 | 38.9 | 35.3 | 38.2 |
|     | 3 | 62.7 | 48.8 | 59.9 | 36.4 | 44.1 | 52.6 | 44.3 | |
|     | AVG.($e$) | 50.1 | 40.2 | 49.9 | 31.6 | 38.1 | 42.7 | | |
| LMPNN | 1 | 20.7 | 29.8 | 33.3 | 13.4 | 16.5 | 21.8 | 19.8 | |
|       | 2 | 63.5 | 35.4 | 43.3 | 27.0 | 30.2 | 37.6 | 32.3 | 35.1 |
|       | 3 | 80.8 | 50.7 | 56.0 | 33.6 | 39.2 | 47.6 | 40.7 | |
|       | AVG.($e$) | 57.4 | 41.5 | 46.7 | 29.4 | 33.6 | 40.0 | | |

Table 8: HIT@10 scores(%) of three different types for answering queries with two free variables on FB15k. The constant number is fixed to be two. The notation of $e$, SDAG, Multi, and Cyclic is the same as Table 2.

| Model | HIT@10 Type | $e=0$ SDAG | $e=0$ Multi | $e=1$ SDAG | $e=1$ Multi | $e=1$ Cyclic | $e=2$ SDAG | $e=2$ Multi | $e=2$ Cyclic | AVG. |
|-------|-------------|------|------|------|------|------|------|------|------|------|
| BetaE | Marginal | 76.9 | 77.2 | 68.9 | 69.3 | 75.1 | 55.0 | 57.4 | 73.6 | 63.6 |
|       | Multiply | 41.7 | 41.6 | 31.7 | 31.0 | 38.7 | 25.2 | 25.9 | 36.1 | 29.7 |
|       | Joint | 11.6 | 13.7 | 8.7 | 8.6 | 17.8 | 4.9 | 5.4 | 14.3 | 8.4 |
| LogicE | Marginal | 82.9 | 80.9 | 73.6 | 72.9 | 76.6 | 58.9 | 60.7 | 75.7 | 66.9 |
|        | Multiply | 47.5 | 45.0 | 36.3 | 34.1 | 40.4 | 28.5 | 29.0 | 38.0 | 32.7 |
|        | Joint | 12.7 | 13.9 | 10.0 | 9.9 | 19.2 | 6.1 | 6.5 | 15.9 | 9.6 |
| ConE | Marginal | 84.1 | 84.8 | 76.5 | 76.3 | 81.4 | 61.8 | 63.8 | 79.7 | 70.2 |
|      | Multiply | 48.7 | 48.1 | 37.7 | 35.9 | 44.2 | 29.9 | 30.4 | 41.4 | 34.6 |
|      | Joint | 14.2 | 15.6 | 10.3 | 10.4 | 20.6 | 6.2 | 6.6 | 16.9 | 10.1 |
| CQD | Marginal | 73.8 | 76.8 | 69.0 | 71.9 | 76.3 | 51.1 | 54.4 | 77.0 | 62.9 |
|     | Multiply | 45.0 | 46.6 | 37.4 | 36.9 | 43.9 | 28.1 | 29.2 | 41.9 | 34.0 |
|     | Joint | 17.1 | 19.0 | 13.1 | 13.0 | 20.6 | 7.7 | 8.6 | 18.1 | 11.9 |
| LMPNN | Marginal | 89.2 | 80.1 | 80.3 | 78.2 | 84.2 | 65.6 | 63.7 | 80.2 | 71.3 |
|       | Multiply | 56.6 | 50.5 | 45.7 | 42.4 | 49.0 | 37.6 | 34.8 | 44.6 | 39.7 |
|       | Joint | 18.9 | 17.2 | 12.9 | 12.4 | 22.4 | 8.0 | 7.5 | 16.9 | 11.2 |

Table 9: HIT@10 scores(%) of three different types for answering queries with two free variables on NELL. The constant number is fixed to be two. The notation of $e$, SDAG, Multi, and Cyclic is the same as Table 2.

| Model | HIT@10 Type | $e = 0$ | | $e = 1$ | | | $e = 2$ | | | AVG. |
|---|---|---|---|---|---|---|---|---|---|---|
| | | SDAG | Multi | SDAG | Multi | Cyclic | SDAG | Multi | Cyclic | |
| BetaE | Marginal | 81.3 | 95.9 | 72.8 | 85.5 | 79.9 | 57.2 | 66.7 | 77.0 | 71.2 |
| | Multiply | 48.2 | 56.7 | 41.3 | 46.1 | 47.6 | 33.1 | 36.5 | 42.9 | 39.6 |
| | Joint | 19.2 | 31.8 | 21.2 | 26.5 | 21.7 | 13.8 | 17.5 | 18.5 | 18.8 |
| LogicE | Marginal | 87.1 | 99.8 | 81.0 | 91.8 | 83.2 | 65.7 | 74.0 | 81.0 | 77.7 |
| | Multiply | 52.5 | 60.3 | 47.6 | 51.7 | 50.2 | 39.4 | 42.6 | 46.0 | 44.8 |
| | Joint | 21.1 | 32.8 | 25.4 | 30.5 | 23.3 | 18.0 | 21.5 | 20.5 | 22.3 |
| ConE | Marginal | 82.6 | 96.4 | 76.0 | 87.8 | 88.1 | 60.0 | 69.3 | 83.0 | 74.7 |
| | Multiply | 48.7 | 56.9 | 41.9 | 46.3 | 52.2 | 34.5 | 38.1 | 47.7 | 41.7 |
| | Joint | 17.0 | 30.9 | 19.3 | 25.0 | 24.9 | 12.9 | 17.2 | 20.3 | 18.8 |
| CQD | Marginal | 79.5 | 96.3 | 83.2 | 92.2 | 83.5 | 65.8 | 75.7 | 84.8 | 79.4 |
| | Multiply | 49.2 | 57.8 | 51.1 | 53.1 | 51.4 | 40.6 | 45.1 | 50.6 | 47.4 |
| | Joint | 23.0 | 38.0 | 29.7 | 34.2 | 26.4 | 21.4 | 25.4 | 24.0 | 26.0 |
| LMPNN | Marginal | 88.5 | 96.6 | 81.5 | 90.9 | 85.3 | 65.0 | 70.7 | 83.1 | 76.7 |
| | Multiply | 55.7 | 62.4 | 50.3 | 53.3 | 54.0 | 40.8 | 42.6 | 50.3 | 46.5 |
| | Joint | 23.4 | 36.4 | 25.5 | 29.4 | 24.0 | 16.6 | 19.7 | 21.5 | 21.5 |

Table 10: HIT@10 scores(%) of three different types for answering queries with two free variables on FB15k-237. The constant number is fixed to be one. The notation of $e$, SDAG, Multi, and Cyclic is the same as Table 2.

| Model | HIT@10 Type | $e = 0$ | | $e = 1$ | | | $e = 2$ | | | AVG. |
|---|---|---|---|---|---|---|---|---|---|---|
| | | SDAG | Multi | SDAG | Multi | Cyclic | SDAG | Multi | Cyclic | |
| BetaE | Marginal | 37.5 | 29.7 | 33.4 | 28.1 | 35.6 | 30.0 | 25.9 | 41.2 | 31.2 |
| | Multiply | 18.9 | 13.7 | 15.3 | 10.3 | 15.2 | 17.7 | 13.3 | 17.2 | 14.3 |
| | Joint | 0.9 | 1.1 | 1.4 | 0.9 | 3.3 | 1.1 | 0.9 | 3.9 | 1.7 |
| LogicE | Marginal | 40.6 | 30.7 | 36.0 | 29.1 | 34.6 | 29.8 | 25.3 | 41.5 | 31.4 |
| | Multiply | 21.1 | 14.3 | 17.2 | 10.9 | 16.3 | 17.8 | 13.3 | 17.5 | 14.7 |
| | Joint | 1.4 | 1.4 | 1.6 | 0.9 | 3.7 | 1.4 | 1.0 | 4.3 | 1.9 |
| ConE | Marginal | 40.8 | 32.4 | 37.3 | 30.4 | 40.7 | 31.1 | 26.9 | 45.0 | 33.5 |
| | Multiply | 22.1 | 15.2 | 18.4 | 11.7 | 19.3 | 18.5 | 14.8 | 20.9 | 16.5 |
| | Joint | 1.4 | 1.0 | 1.7 | 1.0 | 4.3 | 1.4 | 1.0 | 4.4 | 2.0 |
| CQD | Marginal | 73.8 | 76.8 | 69.0 | 71.9 | 76.3 | 51.1 | 54.4 | 77.0 | 62.9 |
| | Multiply | 23.3 | 9.1 | 18.5 | 9.2 | 16.2 | 14.6 | 9.2 | 19.1 | 12.9 |
| | Joint | 1.5 | 0.6 | 2.0 | 1.1 | 3.4 | 1.5 | 0.9 | 4.4 | 1.9 |
| LMPNN | Marginal | 39.0 | 27.6 | 40.0 | 29.5 | 39.3 | 30.6 | 24.8 | 42.7 | 32.0 |
| | Multiply | 25.1 | 13.9 | 24.3 | 13.3 | 21.6 | 20.0 | 14.0 | 21.1 | 17.1 |
| | Joint | 1.6 | 1.3 | 2.5 | 1.3 | 3.9 | 1.5 | 1.0 | 4.0 | 2.0 |

Table 11: HIT@10 scores(%) of two different types for answering queries with two free variables on FB15k-237(including queries without the marginal hard answer). The constant number is fixed to be two. The notation of $e$, SDAG, Multi, and Cyclic is the same as Table 2.

| Model | HIT@10 Type | $e = 0$ SDAG | Multi | $e = 1$ SDAG | Multi | Cyclic | $e = 2$ SDAG | Multi | Cyclic | AVG. |
|-------|-------------|------|-------|------|-------|--------|------|-------|--------|------|
| BetaE | Multiply | 29.1 | 29.1 | 18.3 | 37.5 | 10.4 | 28.0 | 93.6 | 74.6 | 24.1 |
|       | Joint    | 2.1  | 2.2  | 1.7  | 3.0  | 2.4  | 1.8  | 5.8  | 14.2 | 4.6  |
| LogicE | Multiply | 31.6 | 32.9 | 19.8 | 39.6 | 10.9 | 28.7 | 96.3 | 73.8 | 25.4 |
|        | Joint   | 2.6  | 2.5  | 2.1  | 3.1  | 2.5  | 2.2  | 6.4  | 15.6 | 5.0  |
| ConE | Multiply | 32.6 | 31.9 | 20.5 | 41.0 | 12.6 | 29.0 | 99.7 | 86.8 | 27.0 |
|      | Joint     | 3.0  | 2.1  | 1.9  | 3.3  | 2.7  | 2.2  | 6.6  | 16.8 | 5.4  |
| CQD | Multiply | 34.5 | 23.4 | 22.3 | 36.8 | 10.6 | 26.4 | 75.3 | 77.3 | 25.6 |
|     | Joint      | 2.9  | 1.4  | 2.1  | 3.3  | 2.3  | 2.0  | 5.0  | 15.0 | 5.6  |
| LMPNN | Multiply | 36.8 | 29.3 | 27.5 | 45.8 | 13.9 | 31.2 | 97.0 | 86.5 | 27.9 |
|       | Joint    | 2.7  | 2.2  | 2.7  | 3.9  | 2.5  | 2.1  | 5.8  | 14.6 | 5.0  |
| FIT | Multiply | 41.5 | 44.4 | 28.9 | 56.8 | 10.2 | 39.4 | 139.7 | 100.3 | 35.0 |
|     | Joint      | 2.4  | 2.3  | 2.1  | 3.4  | 1.6  | 2.2  | 7.4  | 15.4 | 5.9  |

Table 12: HIT@10 scores(%) of two different types for answering queries with two free variables on FB15k(including queries without the marginal hard answer). The constant number is fixed to be two. The notation of $e$, SDAG, Multi, and Cyclic is the same as Table 2.

| Model | HIT@10 Type | $e = 0$ SDAG | Multi | $e = 1$ SDAG | Multi | Cyclic | $e = 2$ SDAG | Multi | Cyclic | AVG. |
|-------|-------------|------|-------|------|-------|--------|------|-------|--------|------|
| BetaE | Multiply | 42.1 | 57.2 | 26.5 | 66.5 | 15.5 | 34.6 | 134.9 | 100.0 | 35.0 |
|       | Joint    | 6.6  | 9.4  | 4.5  | 10.2 | 4.6  | 4.3  | 16.7  | 26.0  | 9.2  |
| LogicE | Multiply | 48.2 | 65.6 | 31.0 | 71.6 | 16.8 | 37.8 | 143.9 | 105.8 | 38.1 |
|        | Joint   | 7.5  | 11.2 | 5.6  | 12.5 | 5.3  | 5.6  | 20.4  | 28.5  | 10.5 |
| ConE | Multiply | 50.2 | 72.2 | 32.8 | 74.6 | 18.3 | 38.3 | 149.3 | 114.3 | 40.4 |
|      | Joint     | 6.8  | 10.0 | 5.2  | 12.5 | 5.5  | 5.2  | 19.4  | 30.4  | 11.0 |
| CQD | Multiply | 48.1 | 55.9 | 31.9 | 69.0 | 15.8 | 29.5 | 93.5  | 103.2 | 37.6 |
|     | Joint      | 9.4  | 11.4 | 6.6  | 14.8 | 4.8  | 5.5  | 17.5  | 27.2  | 12.0 |
| LMPNN | Multiply | 58.4 | 79.5 | 43.1 | 94.6 | 21.3 | 40.9 | 146.2 | 135.9 | 45.0 |
|       | Joint    | 8.6  | 12.9 | 6.8  | 15.6 | 6.2  | 5.4  | 19.3  | 31.7  | 11.6 |

Table 13: HIT@10 scores(%) of two different types for answering queries with two free variables on NELL(including queries without the marginal hard answer). The constant number is fixed to be two. The notation of $e$, SDAG, Multi, and Cyclic is the same as Table 2.

| Model | HIT@10 Type | $e = 0$ SDAG | Multi | $e = 1$ SDAG | Multi | Cyclic | $e = 2$ SDAG | Multi | Cyclic | AVG. |
|-------|-------------|------|-------|------|-------|--------|------|-------|--------|------|
| BetaE | Multiply | 21.2 | 47.3 | 22.0 | 51.9 | 14.7 | 24.1 | 80.5 | 79.7 | 33.4 |
|       | Joint    | 4.2  | 19.6 | 6.8  | 19.1 | 5.1  | 6.8  | 26.7 | 24.0 | 14.1 |
| LogicE | Multiply | 26.6 | 52.8 | 28.8 | 63.4 | 16.0 | 32.8 | 103.1 | 88.5 | 38.9 |
|        | Joint   | 3.8  | 21.5 | 9.7  | 26.0 | 5.9  | 11.5 | 36.9  | 27.3 | 16.5 |
| ConE | Multiply | 25.3 | 51.4 | 23.9 | 53.9 | 16.9 | 27.3 | 90.7 | 90.6 | 36.7 |
|      | Joint     | 3.4  | 20.2 | 6.4  | 17.0 | 6.1  | 7.2  | 27.0 | 27.1 | 14.2 |
| CQD | Multiply | 30.3 | 48.9 | 30.6 | 64.3 | 15.9 | 33.1 | 88.9 | 91.2 | 40.9 |
|     | Joint      | 4.4  | 21.9 | 9.8  | 27.5 | 5.6  | 12.0 | 37.6 | 28.1 | 18.0 |
| LMPNN | Multiply | 33.4 | 58.3 | 33.7 | 65.3 | 19.4 | 30.7 | 85.1 | 105.0 | 41.8 |
|       | Joint    | 4.4  | 23.7 | 10.0 | 21.9 | 5.8  | 8.2  | 23.2 | 28.8  | 15.7 |

