# OpenReview forum: "${\rm EFO}_k$-CQA: Towards Knowledge Graph Complex Query Answering beyond Set Operation"
_ICLR.cc/2024/Conference — Submitted to ICLR 2024_

### Official Review · Reviewer_LSNr · 2023-10-31

**Soundness:** 3 good
**Presentation:** 2 fair
**Contribution:** 2 fair
**Rating:** 5
**Confidence:** 4

**Summary:**

The authors propose a framework for generating existential first-order
queries over knowledge graphs, which considers different parameters
that make queries harder or easier to evaluate (for example, being
either graph-shaped or tree-shaped). Besides, they use this framework
to compare existing complex query answering models, shedding light on
the sources of the hardness of existential first-order queries over
knowledge graphs.

**Strengths:**

* The development of methods to compare complex query answering models
  over knowledge graphs is important. This paper contributes by
  proposing a comprehensive approach to generate existential
  first-order queries over knowledge graphs, significantly expanding
  the benchmarks for such comparisons.

**Weaknesses:**

* The presentation of the paper needs to be improved.

* The treatment of parameters upon which the difficulty of conjunctive
  query evaluation depends is somewhat superficial. Much research on
  this topic has been conducted in databases, but the authors do not
  take this into consideration.

**Questions:**

(1) The presentation of the paper needs to be improved. In particular,
some definitions need to be clarified.

- Definition 4: Consider the formula $P(x) \wedge \exists x R(x,y)$. In
  this case, both $x$ and $y$ are free variables, although only $y$ is
  considered free according to the definition in the paper. This may
  seem like a minor point, but the correct definition of this notion
  is necessary when considering a logic with a bounded number of
  variables, which is a standard way to construct fragments of
  first-order logic with tractable query evaluation.

- Paragraph after Definition 9: The authors claim the following:
  "... the inference of existential formulas is easier than solving
  CSP instances since the existential variables do not need to be kept
  track of". I do not understand this claim, as the complexity of CSP
  solving and conjunctive query evaluation is the same (both problems
  are NP-complete).

- Definition 7: What does it mean that "$\phi(a_1, ..., a_k)$ is True"?
  Where is this query evaluated? Do you assume a fixed knowledge graph
  over which all queries are evaluated?

- Definition 7: How is the semantics of negation defined under OWA? Do
  you consider a certain answer semantics over some possible worlds
  (otherwise no negative atoms can be inferred)? The authors use the
  term "$\phi(a_1, ..., a_k)$ is True" without defining it.

- Definition 12: For an abstract query graph G, a grounding is a
  function I that maps G into a query graph. Do you impose any
  restrictions on this mapping? For example, could two distinct nodes
  with the type "Free variable" be mapped to the same variable? This
  is relevant for the bottom-left graph in Figure 2, which is claimed
  to violate Assumption 14 (this is not true if you can map both
  yellow nodes to the same variable).

- Paragraph after Assumption 15: The authors mention the following
  "Assumption 15 treats negation separately because of the fact that
  for any KG, any relation r in R, there is |{ (h,t) | h,t in E,
  (h,r,t) in KG}| << E^2". I do not understand this notation. On the
  left-hand side of <<, you count a number of tuples, while on the
  right-hand side, you are considering the cross product of a set with
  itself. Isn't this a comparison of a number with a set?


(2) The treatment of parameters upon which the difficulty of
conjunctive query evaluation depends is somewhat superficial. In fact,
the authors have made a contribution by lifting the restriction that
conjunctive queries must be tree-shaped and by considering an approach
to generate general graph-shaped queries. However, between trees and
general graphs, there exists a large number of structures that are
defined in terms of parameters which have been studied precisely to
analyze the complexity of query evaluation. Most notably, the authors
could have considered the notions of treewidth and hypertree width,
which are explained in the following references:

Jörg Flum, Martin Grohe: Parameterized Complexity Theory. Texts in
Theoretical Computer Science. An EATCS Series, Springer 2006.

Georg Gottlob, Gianluigi Greco, Nicola Leone, Francesco Scarcello:
Hypertree Decompositions: Questions and Answers. PODS 2016: 57-74

Moreover, the following book provides a detailed view of fast
conjunctive query evaluation:

https://github.com/pdm-book/community


(3) Why do you need to develop your own algorithm to compute the
answers to an existential formula, as described in Section 4.3? Why
can't you leverage the substantial body of work and existing
implementations for answering first-order queries?

---

> ### Author Response · Authors · 2023-11-19
>
> Thank you for your valuable feedback. We highly value your expertise in logic and databases, and will improve our paper, including more rigorous definitions, correcting typos in presentation, and citing related previous research according to your suggestions. However, there are also some misunderstandings we would like to clarify:
>
> > Definition 4: Consider the formula…. This may seem like a minor point, but the correct definition of this notion is necessary when considering a logic with a bounded number of variables, which is a standard way to construct fragments of first-order logic with tractable query evaluation.
>
> We appreciate your suggestions. We presume that all variables are named differently as it is the standard way in first order logic without clear explanation. We have corrected this point to make our discussion more clear.
>
> > Paragraph after Definition 9: The authors claim the following: "... the inference of existential formulas is easier than solving CSP instances since the existential variables do not need to be kept track of". I do not understand this claim, as the complexity of CSP solving and conjunctive query evaluation is the same (both problems are NP-complete).
>
> Your understanding is right and both solving  CSP and conjunctive query evaluation are NP-complete. However, our claim is about the polynomial time reduction itself. Moreover, this polynomial reduction is very powerful in a smaller n (<10) like in our cases. In practice, our inference techniques are 100 times faster than CSP solvers. Certainly, both problems are NP-complete and it doesn’t conflict.
>
> > Definition 7: What does it mean that " is True"? Where is this query evaluated? Do you assume a fixed knowledge graph over which all queries are evaluated?
>
> > Definition 7: How is the semantics of negation defined under OWA? Do you consider a certain answer semantics over some possible worlds (otherwise no negative atoms can be inferred)? The authors use the term " is True" without defining it.
>
> Thank you for your suggestions, since our semantics is standard and it is a common practice[1] to state in a intuitive way, we didn’t give a rigorous definition of the semantics of the query. Regarding the evaluation and OWA, the underlying knowledge graph KG is taken as ground truth to evaluate the query. Regarding the negation, any unobserved triple is considered negated. We will include this part of definition in the preliminary.
>
>
> > Definition 12: For an abstract query graph G, a grounding is a function I that maps G into a query graph. Do you impose any restrictions on this mapping?.....
>
> We are extremely sorry for this typo, causing confusion in the reading, as stated in the answer regarding Definition 4, we need to assume all nodes in the abstract query graph G should be named differently. We have corrected this typo thanks to your advice.
>
> > Paragraph after Assumption 15: The authors mention the following "Assumption 15 treats negation separately because of the fact that for any KG, any relation r in R, there is |{ (h,t) | h,t in E, (h,r,t) in KG}| << E^2".....
> Sorry for this Typo, it should be |\mathcal{E}|^2, we have corrected it and thank you for your observation.
>
> >  In fact, the authors have made a contribution by lifting the restriction that conjunctive queries must be tree-shaped and by considering an approach to generate general graph-shaped queries. ….Most notably, the authors could have considered the notions of treewidth and hypertree width…
>
>
> We really appreciate your suggestion. It is really important and necessary to include more parameters like treewidth and is also a common practice in the discussion of traditional database/CSP communities. However, we want to point out that we have made the assumption in Section 5, the footnote on the 8th page that the number of edge of query graph is no more than the number of nodes, therefore, the treewidth is either 2 or 1, making it pointless for discussion. We will leave this to future work to include even more complicated query graph.
>
> > Why do you need to develop your own algorithm to compute the answers to an existential formula, as described in Section 4.3? Why can't you leverage the substantial body of work and existing implementations for answering first-order queries?
>
>
> Thank you for your suggestions. Firstly, our implementation is consistent with existing implementations and we have already double-checked its correctness. Additionally, we will cite more existing work for clarification and we reckon that our own algorithm in Section 4.3 was not a theoretical innovation but rather a convenient implementation that fits into our framework well.
>
> [1] Ren H, Leskovec J. Beta embeddings for multi-hop logical reasoning in knowledge graphs[J]. Advances in Neural Information Processing Systems, 2020, 33: 19716-19726.

---

> > ### Author Response · Authors · 2023-11-21
> > **Looking forward to reply**
> >
> > Dear reviewer, we wonder whether our rebuttal addresses your concern and whether there are any further suggestions or questions. We are looking forward to your reply.

---

> > > ### Author Response · Authors · 2023-11-22
> > > **Looking forward to reply**
> > >
> > > Dear reviewer, we wonder whether our rebuttal addresses your concern and whether there are any further suggestions or questions. We are looking forward to your reply.

---

### Official Review · Reviewer_d2wZ · 2023-10-31

**Soundness:** 4 excellent
**Presentation:** 4 excellent
**Contribution:** 3 good
**Rating:** 3
**Confidence:** 3

**Summary:**

This paper presents a  new framework for studying complex query answering. This framework covers a bigger family of Existential First Order (EFO) queries while previous ones only cover a subset of EFO queries. Besides, the paper introduces a new datasets that contains 741 types of query. The generated queries are several guaranteed to have high quality based on several rules. Finally, the paper implements the entire pipeline for query generation, answer sampling, model training and inference, and evaluation. The paper also includes some evaluation results of existing methods on this benchmark.

**Strengths:**

Strengths:
1) As far as I know, this framework covers the bigest family of EFO queries. The task is much more challenging than that of previous benchmark as they only consider a small subset of the EFO queries. I believe that this framework has enough impact in the knowledge graph reasoning community.
2) Also, when desiging the benchmark, the authors consider both combinatorial hardness and structural hardness
3) The paper introduces a comprehensive benchmark dataset consisting of 741 types of query with guaranteed quality.

**Weaknesses:**

1) The authors discussed some theoretical properties of EFO(k) queries, but did not provide enough insights on how to design a CQA model for the new types of queries that satisfy these properties. It would significantly increase the impact of the paper if the authors could explictly include such a section.
2) It is not clear how the previous model like BetaE/LogicE/ConE etc. are extended to handle cyclic and multigraph queries. These models are known to be able to only handle tree-form queries, as far as I know.

**Post-rebuttal:** After reading the authors' rebuttal and the other reviews, it seems that there are some key limitations that the authors did not address during the rebuttal. Also, there are no methodological contributions or insights for designing new CQA method except a new dataset in this paper. Hence, I downgrade my rating.

**Questions:**

In Fig 3, the authors used some figures from public media. I am not sure whether the authors have the license or need to ask for a license.

---

> ### Author Response · Authors · 2023-11-19
>
> Thank you for your kind feedback. We are honored by your approval of the completeness of our dataset.. We firmly believe that our proposed dataset will contribute to the advancement and development of new models in this field and here is our rebuttal:
>
>
> > The authors discussed some theoretical properties of EFO(k) queries, but did not provide enough insights on how to design a CQA model for the new types of queries that satisfy these properties. It would significantly increase the impact of the paper if the authors could explictly include such a section.
>
> We would like to express our gratitude for your helpful reminder regarding the details of extending previous models to answer queries beyond tree structures. In fact, we believe this is a general direction in the database community, and we have noted very recent research trying to filling this gap[1], we will provide this discussion in our paper. Actually, our work provides some general extension of the previous model like ConE which can only deal with tree-form query initially. And we show that more general methods like LMPNN suffers from the biased selection of training data severely. However, we have not designed any new methods since this is a dataset and benchmark paper, and we strive to test the generalizability of existing methods, providing corresponding discussion. We will leave the more sophisticated design of CQA models in the future but we will surely include more discussions of it thanks to your advice.
>
> > It is not clear how the previous model like BetaE/LogicE/ConE etc. are extended to handle cyclic and multigraph queries. These models are known to be able to only handle tree-form queries, as far as I know.
>
> Thank you for your correct observation, surely these models are known to be able to only handle tree-form queries. In fact, this extension is done by us while ensuring the backward compatibility, this is explained in the Section 4.4 and detailed in Appendix F, and we consider this as one of our contribution of our paper.
>
> [1] Barceló, Pablo, et al. "A neuro-symbolic framework for answering conjunctive queries." arXiv preprint arXiv:2310.04598 (2023).

---

> > ### Author Response · Authors · 2023-11-21
> > **Looking forward to reply**
> >
> > Dear reviewer, we wonder whether our rebuttal addresses your concern and whether there are any further suggestions or questions. We are looking forward to your reply. Also, we would like to add that the figure in social media can be used as long as we give its source.

---

> > > ### Author Response · Authors · 2023-11-22
> > > **Looking forward to reply**
> > >
> > > Dear reviewer, we wonder whether our rebuttal addresses your concern and whether there are any further suggestions or questions. We are looking forward to your reply

---

### Official Review · Reviewer_wF2V · 2023-10-31

**Soundness:** 2 fair
**Presentation:** 2 fair
**Contribution:** 2 fair
**Rating:** 3
**Confidence:** 3

**Summary:**

The paper addresses Complex QA (space of Existential First Order Queries) on a Knowledge Graph. The question is assumed to be represented as a query graph (for first order logic e.g. "Find a city that is located in Europe and is the capital of a country that has not held the Olympics") with associated Answer Set for the free variables of that graph, from the KG, based on previous work.

It proposes a query graph sampling approach to generate graphs, and solve them as constraint satisfaction problems on the KG. These graph-answer pairs are provided as a dataset of 741 query types in first order logic. This I believe is used to train different learning-based methods from prior works, on the query graph to infer the answer set. The evaluation provides the Hit@10 scores for these.

**Strengths:**

1. Dataset of query graphs with Answer Sets (but the quality of the grounding of the query graphs to the real world concepts is not validated in any way. It is unclear how useful the dataset is, or how meaningful the generated query graphs are in terms of real concepts).

2. Heuristic approach for query graph sampling (abstract graph with fixed number of variables, then sampling entities and relations to fill it).

**Weaknesses:**

1. Dataset of query graphs with Answer Sets built from Query graph sampling - It is not validated whether these produce meaningful query graphs when grounded to real-concepts, and how representative is this dataset for Complex QA on a KG with natural questions.

2. There seem to be several key limitations - How does this help with improving Complex questions answering of natural question datasets e.g. WebQSP, Complex WebQSP. First, the approach here does not examine how natural questions from such datasets can be formulated into query graphs with any automatic methods (that are not manual, which would be critical to use this approach). Secondly, it does not suggest how the dataset can improve results on standard QA datasets like Complex WebQSP or the likes for KGQA.

3. I am not clear about how the learning-based methods are relevant here. The dataset is formed by solving some version of CSP (details lacking entirely of their "own algorithm" in Sec 4.3), and then what is the relevance of learning based methods to embed the query graph and try to infer the answer? And this still does not connect to the complex natural question KGQA methods to produce answers for natural questions from the KG.
- Section 4.3 "we develop our own algorithm following the standard solving technique of CSP, which ensures
consistency conditions in the first step, and do the backtracking to get the final answers in the
second step".

4. Approach presentation and clarity - The whole Framework section (4.1-4.4) in the main paper is limited, and does not at all deliver what the approach is (and even if we read through the Appendix, its not presented well enough to grasp the key points of the approach at high-level but with sufficient detail). The presentation is poor with unnecessary assumptions and elaborations (unrelated to the proposed contributions) listed in the main paper, and much of the methodology in the Appendix.

5. This paper is derivative
- It extends to only 2 free variables in practice (and the extension to multiple free variables, over their prior work Yin et al 2023 seems trivial?).
- The two assumptions they define are extraneous as they themselves suggest this cannot be checked in practice (so the contribution claim that "Our assumption is more systematic than previous ones as shown by the example in Figure 2." is not useful.
- We include the whole family of EFO1 query, many of them can not be represented by operator tree. This is based on prior work directly, so its not a contribution of this paper. In which case the only contribution seems to be the dataset with 2 free variables and the heuristics extended to sample and ground the query graphs and get its Answer set for the dataset. The usefulness and solidity of the dataset is unexplored to be clearly justifiable as useful.
- The heuristics to sample and ground the dataset are not obvious, clear, or validated, or effectively presented.

**Questions:**

See weaknesses.

---

> ### Author Response · Authors · 2023-11-19
>
> Thank you for your valuable feedback, which provides a new perspective from the NLP community. However, we believe there are some misunderstandings about the contribution and importance of the task of complex query answering and our work that we would like to clarify.
>
>
>
> > Dataset of query graphs with Answer Sets built from Query graph sampling - It is not validated whether these produce meaningful query graphs when grounded to real-concepts….
>
> > There seem to be several key limitations - How does this help with improving Complex questions answering of natural question datasets e.g. WebQSP, Complex WebQSP..…
>
> We would like to answer these two questions together, as they seem to make the same presumption that query in formal language (first order logic) is only practical when it can be found in natural language, and the task of complex query answering (CQA) is therefore subordinate to natural language task like Knowledge Graph Question Answering (KGQA). However, this is not the case. We would like to refer you to the first question in our general feedback for the discussion where we refute this presumption and we do not reiterate the context here.
>
> In short, the introduction of complicated queries, such as multi-variable queries, even though hard to find in natural language, is crucial for advancing the development of CQA in the sense of strengthening/benchmarking the combinatorial generalizability, which greatly differs from KGQA which is conducted in natural language.
>
> > I am not clear about how the learning-based methods are relevant here…And this still does not connect to the complex natural question KGQA methods to produce answers for natural questions from the KG….
>
> We thank you for your questions. In fact, there have been many previous learning-base methods that focus on complex query answering, and one of our core contributions of this paper, is to benchmark the generalizability, how well they can handle query with different graph features, like different topology properties that may have not been considered during their model design. Regarding the connection to KGQA, we believe this question are addressed above.
>
>
>
>
> > Approach presentation and clarity - The whole Framework section (4.1-4.4) in the main paper is limited,..not presented well enough to grasp the key points …
> Regarding the issue of presentation, we would like to hear actionable suggestions about the context, both in the main paper and the appendix. Our goal is to explain the intuitive and brief idea in the main paper and provide enough details in the appendix. For example, in the main paper, we provide Figure 3, where each component in the framework section  (4.1-4.4) is clearly illustrated, which can grasp the key point of our framework. In the appendix, we offer additional illustrations in Figure 5, together with the writing in Appendix D.1 and Section 4.1, clearly explains how to enumerate all possible abstract query graphs. Similarly, Appendix F is for Section 4.4 where we provide with the pseudo-code here. We will certainly strive to improve the presentation of our paper, but we would like to hear more actionable suggestions.
>
>
>
> > This paper is derivative…..
>
> Regarding this issue, we would like to refer you to the second question in our general rebuttal. Here, we add some brief additional comments.
>
> >> It extends to only 2 free variables in practice (and the extension to multiple free variables, over their prior work Yin et al 2023 seems trivial?).
>
> As discussed in the general rebuttal, the extension from 1 free variable to 2 is fundramental, while 2 to k is derivative instead.
>
> >> The two assumptions they define are extraneous as they themselves suggest this cannot be checked in practice.
>
> This is a very ** clear misunderstandings**. As we state in Section 3.1, Assumption 13 and 14 apply to **abstract query graph** because putting these assumptions to **grounded query graph** should be avoided in practice. Therefore, we strictly abide by these two assumptions we put forward in practice.
>
> >>  We include the whole family of EFO1 query, many of them can not be represented by operator tree. This is based on prior work directly, so it's not a contribution of this paper.
>
> However, no previous work has discussed how to enumerate the whole family of EFO1 query. Those assumptions in Section 3.1 and 3.2, help us greatly when exploring the combinatorial space and has never been discussed to the best of our knowledge.
>
> >> The heuristics to sample and ground the dataset are not obvious, clear, or validated, or effectively presented.
>
> If your concern is about the natural language,  it has been addressed above. If you concern is about our presentation, we would like to hear more detailed suggestions as we have also discussed above.

---

> > ### Comment · Reviewer_wF2V · 2023-11-19
> > **Further questions**
> >
> > I see the difference of complex query answering vs complex question answering (common in the NLP community) of this paper as acceptable now.
> >
> > Still, once the abstract graph is constructed, the entities sampled to created the grounded graph? How does that latter sampling ensure that the queries make logical sense? And there is no validation of that in the paper, correct?
> >
> > My other question is as follows: The CSP approximation solver (since its NP hard otherwise) is used to find the answer set of free variables in that query graph. When sampling, isn't it already known what entities/concepts from the KG fill those free variables? then why is CSP needed? And if I understand correctly, the learning based approach is just used to learn the CSP solution- because prior works have used learning based solutions? But how is it necessary when CSP solver can just be used instead?

---

> > > ### Author Response · Authors · 2023-11-20
> > >
> > > Dear reviewers, we thank you for your valuable comments and appreciate your understanding of the task of the complex query answering now. We would like to make further clarifications below, to facilitate the discussion, we rearrange the answers:
> > >
> > >
> > >
> > > > My other question is as follows: The CSP approximation solver (since its NP hard otherwise) is used to find the answer set of free variables in that query graph. When sampling, isn't it already known what entities/concepts from the KG fill those free variables? then why is CSP needed? And if I understand correctly, the learning based approach is just used to learn the CSP solution- because prior works have used learning based solutions? But how is it necessary when CSP solver can just be used instead?
> > >
> > > We thank you for your concerns. However, as stated in the introduction, we have open world assumption (OWA) about the incompleteness of knowledge graph in our problem setting, which is reiterated in the first paragraph in Section 2.1 that the **observed knowledge graph** is only part of the **real knowledge graph**. Therefore, CQA models are always evaluated by how well they can infer those “hard answers” that can not be found in the observed knowledge graph, which is explained in Section 4.5, we welcome you to check that. On the other hand, a CSP solver can not infer any of those “hard answers”,  and thus CSP solver is only useful when preparing the data when we can use the underlying real knowledge graph rather than just the observed part.
> > >
> > > To summarize, we need learning-based models because we need their generalizability to infer those missing information in the knowledge graph.
> > >
> > >
> > > >  Still, once the abstract graph is constructed, the entities sampled to created the grounded graph? How does that latter sampling ensure that the queries make logical sense? And there is no validation of that in the paper, correct?
> > >
> > >
> > > We thank you for your explicit explanation to your concerns about the sampling/grounding method. In our framework, the abstract query graph defines the syntax of the query to make sure this is a valid query within the EFOk query family (maybe what you call logical sense is valid logical syntax ) and the grounding process gives the query semantics. Then, the grounding should abide by Assumption 15 and 16 in Section 3.2, for example, Assumption 16 follows previous practice to ensure that the query should have not too many answers, and the OWA mentioned above needthe query  at least have one “hard answer”. Therefore, we need to design specific query grounding algorithm, to sample valid query that is suitable for the task of CQA.
> > >
> > >
> > > Again, we appreciate your suggestions, and if you have further questions, we are looking forward to hear that.

---

> > > > ### Author Response · Authors · 2023-11-22
> > > > **Looking forward to reply**
> > > >
> > > > Dear reviewer, we wonder whether our rebuttal addresses your concern and whether there are any further suggestions or questions. We are looking forward to your reply.

---

> > > > > ### Author Response · Authors · 2023-11-22
> > > > > **Looking forward to reply**
> > > > >
> > > > > Dear reviewer, we wonder whether our rebuttal addresses your concern and whether there are any further suggestions or questions. We are looking forward to your reply.

---

### Official Review · Reviewer_Cfvr · 2023-10-31

**Soundness:** 3 good
**Presentation:** 2 fair
**Contribution:** 2 fair
**Rating:** 6
**Confidence:** 3

**Summary:**

The paper proposes a full benchmark for evaluating existential first order queries (in DNF) over knowledge graphs. The benchmark provides a way to generate data, as well as sampling over all possible abstract query types following certain parameters, such as number of variables, as long as they satisfy certain assumptions.
I think the benchmark might prove useful, and gives us a way of sampling all kinds of queries from datasets.
I am somewhat uncomfortable with the benchmark design for queries with several variables, as there is no justification for the choices taken by the authors: why would one try to evaluate queries with more than one free variable by means of architectures designed for unary queries?
Regarding the score, I try to gauge the impact of this benchmark in the community in terms of its difference with standard benchmark of Ren et al. As I see it, the additional power that the authors provide is 1- more unary queries (actually all of them), and 2- support for k-ary queries. I believe this benchmark may impact future contributions in the area of neural query answering, but I don't think just adding more queries would become in an impact that merit publication in ICLR.

**Strengths:**

- benchmark appears to be working, and code is avaliable for future authors.
- benchmark goes beyond what is currently used for papers in the area.
- additional support for answering queries with more than one free variable, even though it is not clear that this is the direction one must go in supporting these queries.

**Weaknesses:**

- benchmark does not include ways of generating new data
- no concern over which queries are more practical than others, results may be altered by queries that would be hard to find in practice.
- limited impact: I consider this as an add on to complement work by Ren et al.

**Questions:**

Please clarify the rationale behind the idea of answering queries with more than 1 free variable by decomposing that queri into several unary pieces. Queries with more than one free variable have a standard interpretation, which is answering tuples, and it is not clear to me that any of the measures would be actually important in practice.

---

> ### Author Response · Authors · 2023-11-19
>
> Thank you for your valuable feedback.  However, we believe there are some misunderstandings about the contribution and importance of the task of complex query answering and our work that we would like to clarify.
>
> > benchmark does not include ways of generating new data.
>
> Thank you for your questions. However, we have produced many new queries with novel query types including both single variable and multi variables. To the best of our knowledge, we surpass all previous work regarding the query types we investigated and the process of sampling data is totally novel.
>
> > no concern over which queries are more practical than others, results may be altered by queries that would be
>
> We thank you for your concerns, however, it seems that you presume query in formal language (first order logic) is only practical when it can be found in natural language. However, this is not the case. We refer you to the first question in our general feedback for the discussion where we refute this presumption. In short, the input of CQA models is in formal language rather than neutral language, and the CQA models aim to be trained on selected query types and combinatorially generalize to unseen complex patterns in first-order logic[1]. In this way, the study of combinatorial generalizability of the CQA models plays a pivotal role in the task of representation learning[2]. Therefore, the introduction of those complicated queries that can hardly be found in natural language has its own merit: benchmarking the combinatorial generalizability of the CQA models.
>
> > limited impact: I consider this as an add on to complement work by Ren et al.
>
> We thank you for your concerns, however, our dataset is a huge extension to the work of Ren[1]. We would like to refer you to the second question in our general feedback. In short, we already explained in the introduction that existing dataset in complex query answering is severely inadequate, because their construction is biased and has not discussed queries with some patterns entirely: multigraph, cyclic graph, multiple free variables, resulting in lack of both combinatorial answers and structural hardness. This huge extension was also mentioned in Section 3.2 and detailed in Appendix D.3, where we provide Figure 6, clearly illustrating how huge our extension is. We welcome you to check that.
>
>
>
>
>
>
>
>
>
> [1] Ren H, Leskovec J. Beta embeddings for multi-hop logical reasoning in knowledge graphs[J]. Advances in Neural Information Processing Systems, 2020, 33: 19716-19726.
>
> [2] Bai, Jiaxin, et al. ‘Sequential Query Encoding for Complex Query Answering on Knowledge Graphs’. Transactions on Machine Learning Research, 2023, https://openreview.net/forum?id=ERqGqZzSu5.

---

> > ### Comment · Reviewer_Cfvr · 2023-11-20
> > **Pplease consider also answering my question**
> >
> > Thanks for the answers. While I think about them and post back, please consider answering my question, about how you treat queries with multiple output variables.

---

> > > ### Author Response · Authors · 2023-11-20
> > >
> > > We are sorry for our negligence and we would like to answer your questions:
> > >
> > > > Please clarify the rationale behind the idea of answering queries with more than 1 free variable by decomposing that queri into several unary pieces. Queries with more than one free variable have a standard interpretation, which is answering tuples, and it is not clear to me that any of the measures would be actually important in practice.
> > >
> > > If you are referring to the construction of our dataset, we abide by Assumption 14, which ensures that a query with more than 1 free variable can not be structurally decomposed into several unary pieces, which ensures the combinatorial hardness we mention in the introduction that each answer should be a tuple and the answer to each free variable are thus correlated. Therefore, the decomposition is strictly avoided when constructing our EFOk-CQA dataset.
> > >
> > > If you are referring to our implementation and evaluation of the previous CQA models, your understanding is correct and we all agree that the ultimate goal is to answer tuples. However, all previous methods **are designed to infer queries with only 1 free variable**, and they fundamentally lack the ability to retrieve combinatorial answers. Therefore, in our implementation, we have to let those models infer the answer of each free variable individually, as explained in Section 4.4 and Appendix F. In our evaluation, **only the joint metric** proposed in Section 4.5 serves the ultimate goal of answering tuples/combinatorial answers, while others are designed as weaker metrics, as a **compromise** when we currently have no method that can retrieve the combinatorial answers. The benchmark result in Table 2 where all methods have very low joint HIT@10 score supports our discussion above and thus needs other easier metrics for the discussion. We believe the marginal and multiply metrics all only serve as milestones towards the ultimate goal, the joint metric.

---

> > > > ### Author Response · Authors · 2023-11-22
> > > > **Looking forward to reply**
> > > >
> > > > Dear reviewer, we wonder whether our rebuttal addresses your concern and whether there are any further suggestions or questions. We are looking forward to your reply.

---

> > > > > ### Author Response · Authors · 2023-11-22
> > > > > **Looking forward to reply**
> > > > >
> > > > > Dear reviewer, we wonder whether our rebuttal addresses your concern and whether there are any further suggestions or questions. We are looking forward to your reply.

---

### Author Response · Authors · 2023-11-19
**General feedback**

We thank the reviewers for the feedback and useful comments. Below, we would like to clarify some important misunderstandings about the contribution and importance of the task of complex query answering (CQA) and our work by putting forward two fundamental questions of our work.


Does the query in formal language (first order logic) only practical when it can be found in natural language ?

No! We would like to emphasize that CQA differs fundamentally from KGQA (Knowledge Graph Question Answering) while both tasks are meaningful. The input of CQA models is in formal language rather than natural language, and the CQA models aim to be trained on selected query types and combinatorially generalize to unseen complex patterns[1]. In this way, the study of combinatorial generalizability of the CQA models plays a pivotal role in this task of representation learning[2]. Therefore, the introduction of those complicated queries that can hardly be found in natural language has its own merit: strengthening/benchmarking the combinatorial generalizability of the CQA models[2].

In contrast, three-hop questions are already the most complex ones in WebQA, and even the query pattern in Complex WebQA is similar to the dataset in [8], which is a very early work in CQA and has been covered by many more recent works like dataset in [1] or our paper. In KGQA, strong combinatorial generalizability is less important than other topics, like natural language understanding.

To summarize, the inclusion of very complex query, is crucial for advancing the development of CQA in the sense of combinatorial generalizability, and greatly differs from KGQA which is conducted in natural language.


Is the extension from query with a single free variable to query with multiple free variables derivative or unimportant?

No! We want to highlight that multiple free variables are substantially different from single free variable and have important potential applications.

The extension itself is not derivative, the reason is the **combinatorial hardness** that has been highlighted in our introduction. To strengthen this purpose, we remove the queries that can be decomposed in Assumption 14. The combinatorial answers put strong challenges in both the inference and the evaluation. In Section 4.5 and Appendix E, we stated that the evaluation protocol for multiple free variables greatly differs from the previous one and has never been discussed, thus we designed three different kinds of metrics for this new challenge. Moreover, the benchmark result in Table 2 shows a very low joint HIT@10 score even though it is a relatively easy metric, exposing the drawbacks that previous methods fail to retrieve combinatorial answers. Overall, both the analytical and the empirical results demonstrate that the extension to two free variables is significant, while extending 2 to general k, on the other hand, is rather derivative.
Regarding the importance of this extension, nowadays, CQA has real-world applications, like fact ranking[3], and explainable recommendation[4]. However, some important practical applications can not be covered by existing dataset in CQA, because their construction is biased and have not discussed queries with multiple free variables entirely. We would like to introduce one example in fraud detection where we need to detect **a group of people** with cyclic money flow for anti-money laundering applications[5], we also note that this finding is also shared by open-source graph database[6,7]. Therefore, our investigation on cyclic queries and queries with more than one free variable can be justified to help develop more versatile CQA models that are suitable for more real-world applications.
We hope that our answer to these two fundamental questions may help with better understanding about the value and contribution of both the CQA task and our paper.


[1] Ren H, Leskovec J. Beta embeddings for multi-hop logical reasoning in knowledge graphs[J]. Advances in Neural Information Processing Systems, 2020, 33: 19716-19726.


[2] Bai, Jiaxin, et al. ‘Sequential Query Encoding for Complex Query Answering on Knowledge Graphs’. Transactions on Machine Learning Research, 2023, https://openreview.net/forum?id=ERqGqZzSu5.


[3] Ren, Hongyu, et al. "Fact Ranking over Large-Scale Knowledge Graphs with Reasoning Embedding Models." Data Engineering: 124.

[4] Syed, Muzamil Hussain, Tran Quoc Bao Huy, and Sun-Tae Chung. "Context-aware explainable recommendation based on domain knowledge graph." Big Data and Cognitive Computing 6.1 (2022): 11.

[5] Priya, Jithin Mathews, et al. "A graph theoretical approach for identifying fraudulent transactions in circular trading." DATA ANALYTICS 2017 (2017): 36.

[6] https://www.nebula-graph.io/

[7] https://www.nebula-graph.io/posts/fraud-detection-using-knowledge-and-graph-database

[8] Hamilton, Will, et al. "Embedding logical queries on knowledge graphs." Advances in neural information processing systems 31 (2018).

---

### Author Response · Authors · 2023-11-22
**Summary of strengths**

We thank the reviewers for the feedback and useful comments. In addition to detailed responses, in this general comment, we would like to provide a summary of identified strengths and changes in the new revision that are marked in blue.

Strengths:


- We cover the biggest family of EFO queries, far beyond all previous work. (Reviewer Cfvr: *benchmark goes beyond what is currently used for papers in the area*, Reviewer wF2V: *Dataset of query graphs with Answer Sets*, Reviewer d2wZ: *As far as I know, this framework covers the bigest family of EFO queries.*, Reviewer LSNr: *significantly expanding the benchmarks for such comparisons*)
- We proposed useful assumptions that help us explore the combinatorial space of existential first order query. (Reviewer wF2V: *Heuristic approach for query graph sampling*, Reviewer d2wZ: *Also, when desiging the benchmark, the authors consider both combinatorial hardness and structural hardness*)

Moreover, we made many revisions across our paper, correcting typos, making the definition more rigorous, and so on, we welcome the reviewer to check that.

---

### Meta-Review · Area_Chair_FA1J · 2023-12-23

**Metareview:**

Overall, the paper presents a potentially valuable benchmark for query evaluation over knowledge graphs, but it requires improvements.  The paper would be strengthened if it provided a more detailed and clear presentation, especially regarding key definitions and methodologies, and validated the practical relevance and grounding of the query graphs.  The present paper, unfortunately, is not strong enough for acceptance in neurips.

**Justification For Why Not Higher Score:**

The readability of the paper should be improved significantly.

**Justification For Why Not Lower Score:**

The paper has the potential for query evaluation of knowledge graphs.

---

### Decision · Program_Chairs · 2024-01-16

Reject